# Molecular basis for activation of lecithin: cholesterol acyltransferase by a compound that increases HDL cholesterol

Kelly A Manthei[1], Shyh-Ming Yang[2], Bolormaa Baljinnyam[2], Louise Chang[1], Alisa Glukhova[1], Wenmin Yuan[3], Lita A Freeman[4], David J Maloney[2], Anna Schwendeman[3], Alan T Remaley[4], Ajit Jadhav[2], John JG Tesmer[5]*

[1]Life Sciences Institute, University of Michigan, Ann Arbor, United States; [2]National Center for Advancing Translational Sciences, National Institutes of Health, Rockville, United States; [3]Department of Pharmaceutical Sciences and Biointerfaces Institute, University of Michigan, Ann Arbor, United States; [4]Lipoprotein Metabolism Section, Cardiovascular-Pulmonary Branch, National Heart, Lung, and Blood Institute, National Institutes of Health, Bethesda, United States; [5]Department of Biological Sciences, Purdue University, Indiana, United States

**Abstract** Lecithin:cholesterol acyltransferase (LCAT) and LCAT-activating compounds are being investigated as treatments for coronary heart disease (CHD) and familial LCAT deficiency (FLD). Herein we report the crystal structure of human LCAT in complex with a potent piperidinylpyrazolopyridine activator and an acyl intermediate-like inhibitor, revealing LCAT in an active conformation. Unlike other LCAT activators, the piperidinylpyrazolopyridine activator binds exclusively to the membrane-binding domain (MBD). Functional studies indicate that the compound does not modulate the affinity of LCAT for HDL, but instead stabilizes residues in the MBD and facilitates channeling of substrates into the active site. By demonstrating that these activators increase the activity of an FLD variant, we show that compounds targeting the MBD have therapeutic potential. Our data better define the substrate binding site of LCAT and pave the way for rational design of LCAT agonists and improved biotherapeutics for augmenting or restoring reverse cholesterol transport in CHD and FLD patients.
DOI: https://doi.org/10.7554/eLife.41604.001

*For correspondence:
jtesmer@purdue.edu

**Competing interests:** The authors declare that no competing interests exist.

## Introduction

Coronary heart disease (CHD) is the leading cause of death in the world and typically develops as the result of atherosclerotic plaque build-up in the arteries. Risk for CHD is inversely related to high-density lipoprotein (HDL) cholesterol (HDL-C) levels in plasma. In reverse cholesterol transport (RCT), HDL receives cholesterol from cholesterol-enriched macrophages, which is then esterified by lecithin:cholesterol acyltransferase (LCAT) bound to HDL. LCAT preferentially catalyzes the transfer of the *sn*-2 acyl group from phosphatidylcholine (lecithin) to cholesterol, creating a cholesteryl ester (CE) that partitions to the hydrophobic core of the HDL particle (*Calabresi et al., 2012*). This process drives the maturation of discoidal pre-β HDL to spherical α-HDL and promotes further cholesterol efflux from arterial plaques (*Glomset, 1968*).

LCAT esterification of cholesterol in HDL is promoted by ApoA-I, the most abundant structural apolipoprotein in HDL (*Fielding et al., 1972*; *Jonas, 2000*). The structural determinants that underlie ApoA-I activation of LCAT are poorly understood, but clues have been provided by a series of crystal structures of LCAT (*Gunawardane et al., 2016*; *Manthei et al., 2017*; *Piper et al., 2015*) and the closely-related lysosomal phospholipase A2 (LPLA2) (*Glukhova et al., 2015*). Both enzymes contain

**eLife digest** Cholesterol is a fatty substance found throughout the body that is essential to our health. However, if too much cholesterol builds up in our blood vessels, it can cause blockages that lead to heart and kidney problems. The body removes excess cholesterol by sending out high-density lipoproteins (HDL) that capture the fatty molecules and carry them to the liver where they are eliminated. The first step in this process requires an enzyme called LCAT, which converts cholesterol into a form that HDL particles can efficiently pack and transport. The enzyme acts by interacting with HDL particles, and chemically joining cholesterol with another compound.

Finding ways to make LCAT perform better and produce more HDL could improve treatments for heart disease. This could be particularly helpful to people with genetic changes that make LCAT defective. Several small molecules that 'dial up' the activity of LCAT have been identified, but how they act on the enzyme is not always well understood.

Manthei et al. therefore set out to determine precisely how one such small activator promotes LCAT function. The experiments involved using a method known as crystallography to look at the structure of LCAT when it is attached to the small molecule. They also evaluated the activity of the enzyme and other aspects of the protein in the presence of the small molecule and HDL particles.

Taken together, the results led Manthei et al. to suggest that the small molecule works by more efficiently bringing into LCAT the materials that this enzyme needs to create the transport-ready form of cholesterol. The small molecule also partially restored the activity of mutant LCAT found in human disease.

This knowledge may help to design more drug-like chemicals to 'boost' the activity of LCAT and prevent heart and kidney disease, especially in people who carry a defective version of the enzyme.
DOI: https://doi.org/10.7554/eLife.41604.002

an α/β-hydrolase domain and two accessory domains referred to as the membrane-binding domain (MBD) and cap domain. The MBD contains hydrophobic residues important for LPLA2 to bind liposomes and for LCAT to bind HDLs. Protruding from the cap domain is an active site lid that has been observed in multiple conformations. In the case of LCAT, crystallographic and hydrogen/deuterium exchange mass spectrometry (HDX MS) studies suggest that the lid blocks the active site in its inactive state, and opens in response to the binding of substrates and, presumably, upon interaction with HDL (*Manthei et al., 2017*). The lid region is also important for HDL-binding (*Cooke et al., 2018*; *Glukhova et al., 2015*; *Manthei et al., 2017*), and thus we hypothesize that activation imposed by ApoA-I involves conformational changes in LCAT that stabilize its lid in an open state that is more competent to bind substrates.

To date, over 90 genetic mutations in LCAT have been described and are responsible for two phenotypes of LCAT deficiency: fish-eye disease (FED), wherein patients retain residual LCAT activity, particularly on apoB-containing lipoproteins, and familial LCAT deficiency (FLD), wherein patients exhibit a total loss of LCAT activity (*Kuivenhoven et al., 1997*; *Rousset et al., 2009*). Both are characterized by low levels of HDL-C and corneal opacities, but FLD presents additional serious symptoms including anemia, proteinuria, and progressive renal disease, the main cause of morbidity and mortality in these patients (*Ahsan et al., 2014*; *Ossoli et al., 2016*; *Rousset et al., 2011*). Novel treatments for raising HDL-C largely based on inhibition of cholesteryl ester transfer protein have failed to protect against CHD in clinical trials (*Kingwell et al., 2014*; *Rader, 2016*). Therefore, there is currently great interest in investigating alternative pathways for modulating HDL metabolism. In particular, the focus has switched from raising HDL-C to developing drugs that increase the beneficial properties of HDL, such as cholesterol efflux, which is enhanced by LCAT (*Czarnecka and Yokoyama, 1996*). New treatments that increase LCAT activity could therefore be beneficial for both FLD and CHD patients.

Recombinant human LCAT (rhLCAT), which raises HDL-C and increases cholesterol efflux, was shown to be safe in a phase I study (*Shamburek et al., 2016b*) and is now in phase II trials for CHD (clinicaltrials.gov, NCT02601560, NCT03578809). This same rhLCAT has also been tested in enzyme replacement therapy for one patient with FLD with encouraging results (*Shamburek et al., 2016a*). However, small molecule activators would be less expensive and easier to administer than a

biotherapeutic. Previously, Amgen identified Compound A (3-(5-(ethylthio)−1,3,4-thiadiazol-2-ylthio) pyrazine-2-carbonitrile)), which binds covalently to Cys31 in the active site of LCAT and increases plasma CE and HDL-C levels in mice and hamsters (*Chen et al., 2012*; *Freeman et al., 2017*; *Kayser et al., 2013*). Other sulfhydryl-reactive compounds based on monocyclic β-lactams have also been shown to activate LCAT (*Freeman et al., 2017*). Although highlighting the promise of LCAT-activating molecules, these compounds are expected to have many off-target effects. Recently, Daii-chi Sankyo reported a new class of reversible small molecule activators that have demonstrated the ability to activate LCAT isolated from human plasma (*Kobayashi et al., 2016*; *Kobayashi et al., 2015a*; *Kobayashi et al., 2015b*; *Onoda et al., 2015*), and increased HDL-C up to 1000-fold when orally administered to cynomolgus monkeys (*Onoda et al., 2015*).

Here we determined the structure of LCAT bound to both a Daiichi Sankyo piperidinylpyrazolo-pyridine activator and isopropyl dodecyl fluorophosphonate (IDFP), a covalent inhibitor that mimics an acylated reaction intermediate, in which the enzyme adopts an active conformation with an open lid. The activator binds in a pocket formed exclusively by the MBD but does not influence affinity of LCAT for HDL. The lid, which contains positions mutated in FLD, undergoes a large conformational change from that observed in inactive LCAT structures. We show that variants of Arg244 within the lid recover acyltransferase activity when treated with a piperidinylpyrazolopyridine activator, highlighting the promise of compounds that target the MBD for many missense FLD variants. Our results thereby provide a better understanding of the key conformational changes that LCAT under-goes during activation, insight into how the enzyme alters its conformation in response to acyl sub-strates, and a rational framework for the design of new small molecule LCAT modulators.

## Results

### Characterization of LCAT activators

We first synthesized and confirmed the ability of three recently reported piperidinylpyrazolopyridine and piperidinylimidazopyridine LCAT activators (*Kobayashi et al., 2015a*; *Onoda et al., 2015*) (compounds 1–3, *Figure 1a*) to activate hydrolysis of 4-methylumbelliferyl palmitate (MUP) by full-length LCAT (*Figure 1b*). All three activated LCAT greater than 2-fold, with $EC_{50}$ values of 160, 280 and 320 nM for 1, 2, and 3, respectively (*Tables 1* and *2*). We also examined the acyltransferase activity of LCAT with dehydroergosterol (DHE) incorporated in peptide-based HDLs in response to compound 2, as it has lower background fluorescence in this assay. We observed that compound 2 activates LCAT 2.8-fold with an $EC_{50}$ of 280 nM (*Table 1*, *Figure 1c*). To gain insight into the mechanism of activation, we determined the $V_{max}$ and $K_m$ values for the DHE assay with and without 5 µM compound 2. The $V_{max}$ increased from 22 to 37 µM DHE-ester $hr^{-1}$, whereas the $K_m$ was not significantly changed (11 µM vs. 6.6 µM with compound 2) (*Figure 1d*).

We next examined the ability of compound 1 to modulate HDL-binding by pre-incubating the compound with LCAT and then monitoring the kinetics of LCAT binding to ApoA-I HDLs with bio-layer interferometry (BLI). There was no change in the $k_{on}$, $k_{off}$, or overall $K_d$ in BLI, and thus the compounds do not appear to act by increasing LCAT affinity for HDL (*Table 3*, *Figure 1e*, *Figure 1—figure supplement 1a*). The activators did however increase the melting temperature ($T_m$) of LCAT ($\Delta T_m$ values of 2.7–5.0 °C), similar to that which occurs upon reaction of LCAT with isopropyl dodecyl fluorophosphonate (IDFP) ($\Delta T_m = 7$ °C) (*Manthei et al., 2017*) (*Figure 1f–g*). A $K_d$ value of 100 ± 14 nM was determined for compound 1 binding to LCAT via microscale thermophoresis (MST) (*Figure 1—figure supplement 1b*).

### Structure of activated LCAT

With the goal of visualizing an active conformation of LCAT, we examined the combined ability of both compound 1 and IDFP to stabilize ΔNΔC-LCAT (residues 21–397), a truncation variant that lacks the dynamic N- and C-termini of the enzyme and thus is more readily crystallized (*Glukhova et al., 2015*; *Gunawardane et al., 2016*; *Manthei et al., 2017*; *Piper et al., 2015*). The ligands had an additive effect ($\Delta T_m$ of 12.7 °C), suggesting that the two ligands have distinct, non-overlapping bind-ing sites (*Figure 1f–g*). Because increased protein stability improves the chances of obtaining crys-tals, ΔNΔC-LCAT incubated with both IDFP and 1 (ΔNΔC-IDFP·1) was thus subjected to crystallization trials. The combined use of these ligands was expected to trap an active conformation

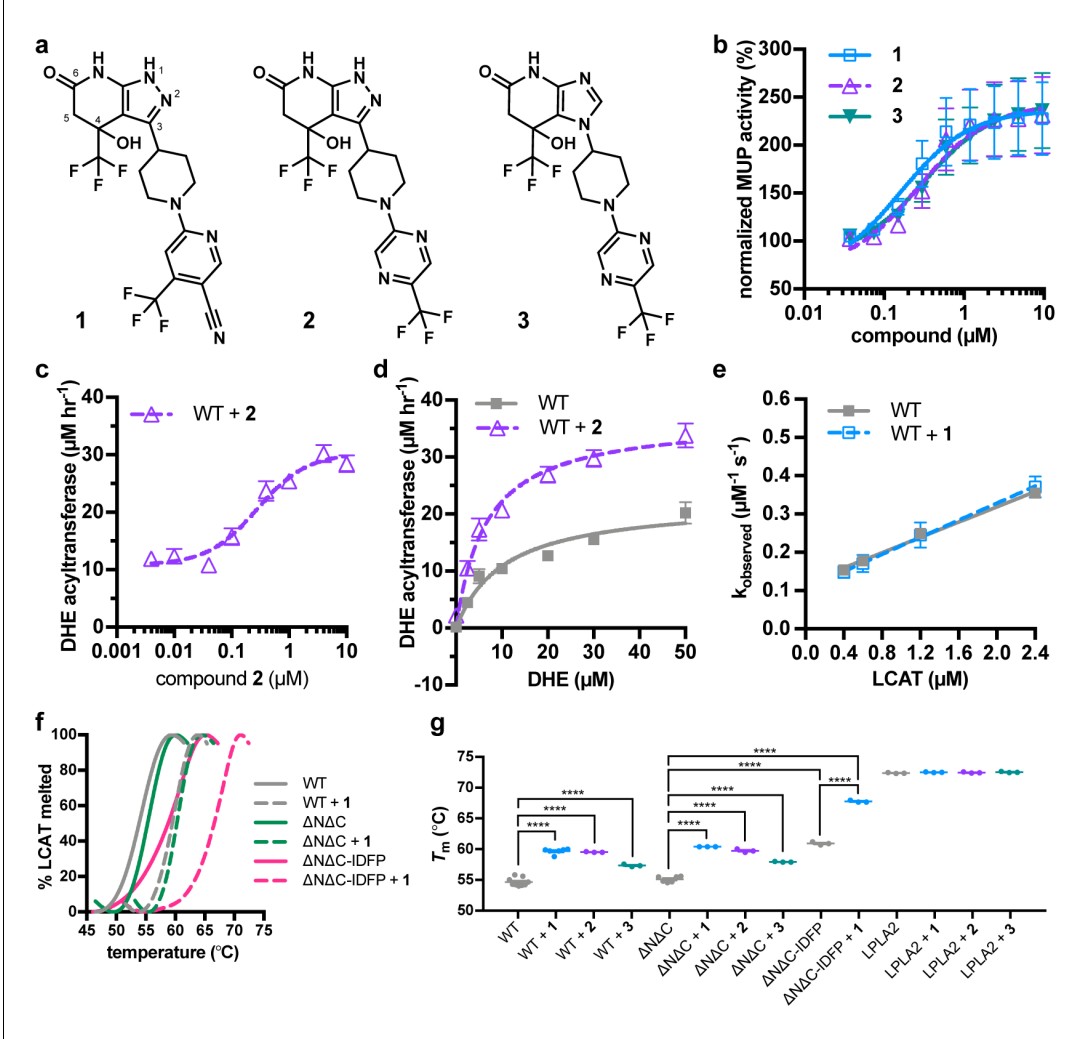

**Figure 1.** Piperidinylpyrazolopyridine and related activators stimulate and stabilize LCAT. (a) Structure of compounds **1** (patent example 95 (**Kobayashi et al., 2015a**)), **2** (patent example 46 (**Kobayashi et al., 2015a**)), and **3** (patent example 3 (**Onoda et al., 2015**)). (b) All three activators stimulate LCAT in a micelle-based MUP assay. Data shown are mean ± s.e.m. from three independent experiments, and data were normalized to basal LCAT activity. (c) Titration of compound **2** (used in this particular assay due to its lower background fluorescence) in the DHE acyltransferase assay. Data shown are mean ± s.e.m. from three independent experiments performed in triplicate. (d) The addition of 5 μM compound **2** stimulates LCAT acyltransferase activity. Data shown are mean ± s.e.m. from three independent experiments performed in triplicate. (e) The addition of 10 μM compound **1** does not affect LCAT binding to HDL as measured with BLI. Plot used to determine $k_{on}$, $k_{off}$, and hence $K_d$. Data are mean ± s.e.m. of three independent experiments. (f) Representative DSF data highlighting the additive increase in $T_m$ induced by combination of **1** and IDFP. Data are normalized from 0% to 100% using the lowest and highest values, respectively. (g) Compounds **1**, **2**, and **3** stabilize WT, ΔNΔC, and ΔNΔC-IDFP LCAT, but not LPLA2. DSF data are mean ± s.e.m. of at least three independent experiments performed in duplicate. ****$p<0.0001$ by one-way analysis of variance followed by Tukey's multiple comparisons post-test. Each protein without ligand was compared to same variant with ligand, and non-significant pairs are not shown. WT compared to ΔNΔC was not significant.

DOI: https://doi.org/10.7554/eLife.41604.003

The following figure supplement is available for figure 1:

**Figure supplement 1.** Effects of LCAT binding to compound **1**.
DOI: https://doi.org/10.7554/eLife.41604.004

**Table 1.** EC$_{50}$ values of LCAT variants in esterase and acyltransferase assays.

| | MUP assay EC$_{50}$ (µM) | | | DHE assay EC$_{50}$ (µM) |
|---|---|---|---|---|
| Variant\Compound | 1 | 2 | 3 | 2 |
| WT | 0.16 ± 0.01 | 0.28 ± 0.04 | 0.32 ± 0.05 | 0.28 ± 0.09 |
| Y51S | 0.59 ± 0.03 | 0.74 ± 0.2 | 1.6 ± 0.4 | ND |
| G71I | >5 | >5 | >5 | ND |
| Y51S/G71I | no effect | no effect | no effect | no effect |
| R244A | 0.13 ± 0.02 | 0.27 ± 0.04 | 0.40 ± 0.03 | 0.76 ± 0.2 |
| R244H | 0.16 ± 0.03 | 0.32 ± 0.03 | 0.47 ± 0.05 | 4.6 ± 2 |

ND = not determined. In the MUP esterase assay, compound was titrated from 0.04 to 9.5 µM, and reactions were performed in triplicate. In the DHE acyl-transferase assay, compound 2 was titrated from 0.004 to 10 µM and reactions were performed three times in triplicate. Values reported are mean ± s.e.m.

DOI: https://doi.org/10.7554/eLife.41604.005

of LCAT. The resulting structure was determined using diffraction data to 3.1 Å spacings (*Figure 2*, *Figure 2—figure supplement 1*, *Table 4*). Crystals could not be obtained without both ligands. There are two protomers of ΔNΔC-IDFP·**1** in the asymmetric unit with a root mean square deviation (RMSD) of 0.35 Å for all Cα atoms, indicating nearly identical conformations (*Krissinel and Henrick, 2004*). Density was observed for residues spanning 21–397 of chain A and 21–395 of chain B, although in both chains a portion of the lid is disordered (239–240 in chain A and 236–242 in chain B).

Strong omit map density is observed for both compound **1** and portions of IDFP (*Figure 2b–d* ). Compound **1** binds in a groove formed by the MBD of each subunit, burying 380 Å$^2$ of accessible surface area of the protein (*Pettersen et al., 2004*) (*Figure 2b–c*). The bicyclic head of **1** binds in a pocket chiefly formed by the b1-b2 loop and a1 and a2 helices (nomenclature as in LPLA2 (*Glukhova et al., 2015*)), including the Cys50-Cys74 disulfide bond (*Figure 2a–c*). Its pyrazole ring donates and accepts a hydrogen bond with the backbone carbonyl and amide of Met49 and Tyr51, respectively, which mandates the hydrogen to be on the 2-position of the ring (*Figure 2—figure supplement 2a*, Compound **1-b**). The C4 hydroxyl donates a hydrogen bond to the side chain of Asp63, and the C6 carbonyl accepts a hydrogen bond from the side chain of Asn78. The C4 trifluoromethyl group is buried against the a1 and a2 helices. Thus, although compound **1** was synthesized as a racemic mixture at the C4 position, the binding site is only compatible with the *R* enantiomer (*Figure 2—figure supplement 2a*, Compound **1-c**). For simplicity, in future descriptions the compound observed in the structure is still referred to as compound **1**. The stereochemical preference is consistent with previous observations that one optical enantiomer of a given activator is typically at least ten-fold more potent than the other (*Kobayashi et al., 2015a*; *Kobayashi et al., 2015b*). The pyrazole moiety packs between the side chain of Tyr51 and the Cys50-Cys74 disulfide. The central piperidine ring of **1** forms van der Waals contacts, but also positions the terminal pyrazine ring of **1** in a hydrophobic cleft formed by the side chains of Met49, Leu68, Pro69, and Leu70 (*Figure 2b*). One edge of the pyrazine moiety also participates in crystal lattice contacts with

**Table 2.** Fold activation for LCAT variants in the MUP esterase assay.

| | Fold activation | | | | | |
|---|---|---|---|---|---|---|
| Variant\Compound | 1 | 2 | 3 | 6 | 8 | 9 |
| WT | 2.3 ± 0.4 | 2.3 ± 0.4 | 2.4 ± 0.4 | no effect | 3.7 ± 0.9 | 1.6 ± 0.2 |
| Y51S | 1.9 ± 0.2 | 1.8 ± 0.1 | 1.9 ± 0.2 | no effect | 2.8 ± 0.6 | 1.1 ± 0.07 |
| G71I | 1.5 ± 0.4 | 1.7 ± 0.2 | 1.5 ± 0.2 | no effect | 1.2 ± 0.1 | 0.96 ± 0.01 |
| Y51S/G71I | 1.3 ± 0.3 | 0.99 ± 0.03 | 1.1 ± 0.06 | no effect | 1.1 ± 0.07 | 0.97 ± 0.003 |
| R244A | 1.7 ± 0.4 | 1.9 ± 0.2 | 1.9 ± 0.2 | no effect | 3.2 ± 0.8 | 1.2 ± 0.1 |
| R244H | 1.6 ± 0.3 | 1.8 ± 0.1 | 1.8 ± 0.1 | no effect | 2.8 ± 0.6 | 1.3 ± 0.06 |

Compound was titrated from 0.04 to 9.5 µM, and reactions were performed in triplicate with values reported as mean ± s.e.m.

DOI: https://doi.org/10.7554/eLife.41604.006

**Table 3.** Effect of LCAT mutations and compound **1** on HDL binding.

| Variant | $k_{on}$ ($s^{-1}$ $\mu M^{-1}$) | $k_{off}$ ($s^{-1}$) | $K_d$ ($\mu M$) |
|---|---|---|---|
| WT | 0.10 ± 0.006 | 0.12 ± 0.008 | 1.2 |
| WT + **1** | 0.11 ± 0.003 | 0.11 ± 0.004 | 1.0 |
| Y51S/G71I | 0.074 ± 0.02 | 0.33 ± 0.03 | 4.5 |
| R244A | 0.069 ± 0.003 | 0.22 ± 0.005 | 3.2 |
| R244A + **1** | 0.017 ± 0.009 | 0.19 ± 0.01 | 11 |
| R244H | 0.022 ± 0.002 | 0.40 ± 0.004 | 18 |
| R244H + **1** | 0.035 ± 0.005 | 0.15 ± 0.007 | 4.3 |

HDLs were attached to streptavidin tips via biotinylated lipid, then dipped into LCAT without or with 10 µM compound **1**. LCAT was titrated from 0.4 to 2.4 µM, $k_{obs}$ was calculated for each concentration and plotted against concentration. Reactions were performed in triplicate and values are reported as mean ± s.e.m.

DOI: https://doi.org/10.7554/eLife.41604.007

residues in the αA-αA' loop (residues 111–119), a region proposed to be involved in cholesterol binding (*Glukhova et al., 2015*; *Manthei et al., 2017*), although these lattice contacts are distinct in

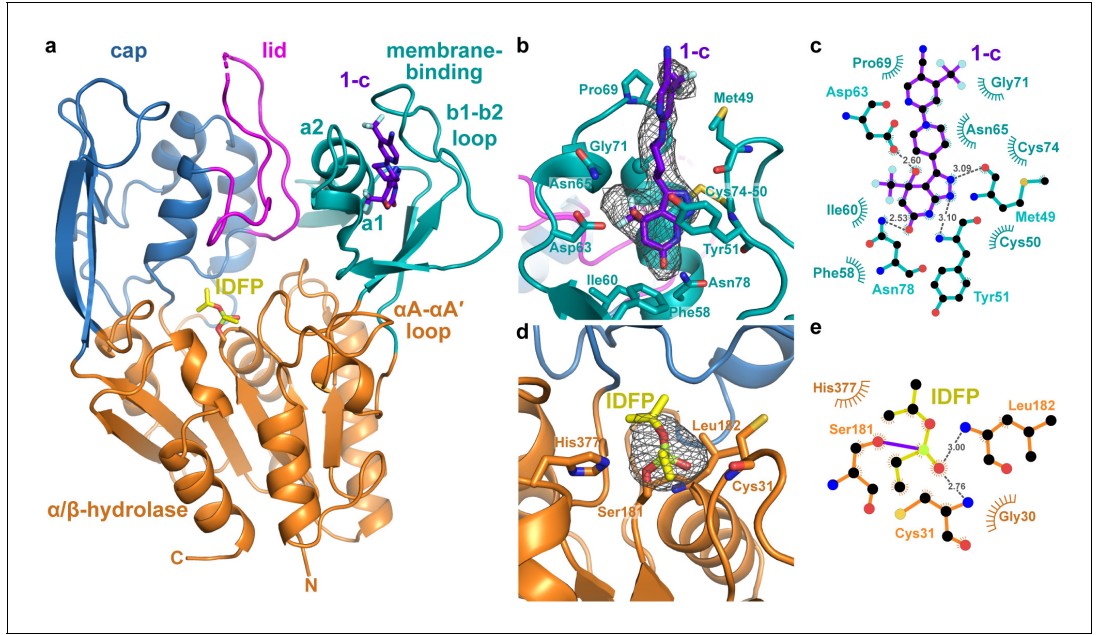

**Figure 2.** Structure of the ΔNΔC-IDFP·**1** complex. (**a**) 3.1 Å X-ray crystal structure highlighting the three domains of LCAT and the binding sites for compound **1-c** (purple) and IDFP (yellow), shown as stick models. The hydrolase domain is shown in orange, cap domain in blue, lid in magenta, and membrane-binding domain (MBD) in teal. (**b**) Closeup of **1-c** bound to the MBD, with $|F_o|$-$|F_c|$ omit map density contoured at 3 σ in gray mesh. (**c**) LigPlot (*Laskowski and Swindells, 2011*) of **1-c** bound to LCAT showing interactions between protein and ligand. Hydrogen bonds are indicated by gray dashed lines with distances in Å. (**d**) IDFP attached to catalytic Ser181, with $|F_o|$-$|F_c|$ omit map density contoured at 3 σ in gray mesh. (**e**) LigPlot of IDFP bound covalently to LCAT at Ser181. The covalent point of attachment is indicated by a purple bond. Protein carbons are colored according to their respective domains or ligands (panel a), whereas nitrogens are blue, oxygens red, sulfurs yellow, and phosphate lime green.

DOI: https://doi.org/10.7554/eLife.41604.008

The following figure supplements are available for figure 2:

**Figure supplement 1.** Electron density maps and crystal packing of the ΔNΔC-IDFP·**1** complex.
DOI: https://doi.org/10.7554/eLife.41604.009
**Figure supplement 2.** Compound structures and numbers.
DOI: https://doi.org/10.7554/eLife.41604.010
**Figure supplement 3.** Asymmetric unit of the ΔNΔC-IDFP·1 crystals and interactions of compound 1-c.
DOI: https://doi.org/10.7554/eLife.41604.011

**Table 4.** Data collection and refinement statistics.

| Data collection | ΔNΔC-IDFP·1 Complex (PDB entry 6MVD) |
|---|---|
| Space group | C2 |
| Cell dimensions | |
| $a$, $b$, $c$ (Å) | 134.5, 106.7, 117.8 |
| α, β, γ (°) | 90.0, 125.5, 90.0 |
| Resolution (Å) | 30.0–3.10 (3.15–3.10)* |
| $R_{merge}$ | 0.115 (≥1) |
| I / σ$_I$ | 11.1 (1.27) |
| Completeness (%) | 98.9 (100.0) |
| Redundancy | 4.2 (4.2) |
| $CC_{1/2}$ | (0.55) |
| **Refinement** | |
| Resolution (Å) | 28.8–3.10 |
| No. reflections | 20,413 |
| $R_{work}$/$R_{free}$ | 19.3/23.9 |
| Number of atoms | 6182 |
| Protein | 5978 |
| Ligand | 183 |
| Water | 20 |
| $B$-factors (Å$^2$) | |
| Overall | 73.6 |
| Protein | 73.2 |
| Ligand | 91.4 |
| Water | 41.1 |
| R.m.s. deviations | |
| Bond lengths (Å) | 0.008 |
| Bond angles (°) | 1.33 |
| Ramachandran statistics | |
| Favored | 93.5 |
| Allowed | 6.0 |
| Outliers | 0.5 |
| MolProbity score | 2.19 |
| Clashscore, all atoms | 4.4 |

*Values in parentheses are for the highest-resolution shell.

DOI: https://doi.org/10.7554/eLife.41604.012

each chain (*Figure 2—figure supplement 3a–b*). This contact may explain why similar crystals could not be obtained with compounds **2** and **3**, which have bulky trifluoromethyl substitutions for the pyrazine cyano group. Notably, the binding site for compound **1** is also occupied in some prior LCAT and LPLA2 crystal structures (*Figure 3a–b*), either by a Phe-Tyr dipeptide of an inhibitory Fab fragment (Fab1) (PDB entries 4XWG, 4XX1, 5BV7) (*Gunawardane et al., 2016*; *Piper et al., 2015*) or by a HEPES molecule in structures of LPLA2 (*Glukhova et al., 2015*), indicating that the MBD in the LCAT/LPLA2 family serves as a robust binding site for diverse chemical matter. Because the 4XWG and 4XX1 structures (referred to as LCAT–Fab1) of LCAT adopt what seems to be an inactive conformation (*Manthei et al., 2017*; *Piper et al., 2015*), a general occupation of the activator binding site however seems insufficient to trigger a global conformational transition in LCAT.

The strongest omit density for IDFP corresponds to its phosphonate head group, which is covalently bound to Ser181 and occupies the oxyanion hole (*Figure 2d–e*). The density is progressively weaker beyond the phosphonate, and the alkyl chain past the C2 carbon is not observed. However, the location of IDFP in our structure and the dynamic nature of the alkyl chain is consistent with results from the LPLA2-IDFP structure (PDB entry 4X91), wherein multiple conformations of bound IDFP suggested two hydrophobic tracks likely used for binding the acyl chains of phospholipid substrates (*Glukhova et al., 2015*) (*Figure 2—figure supplement 3c*). Indeed, there is a similar

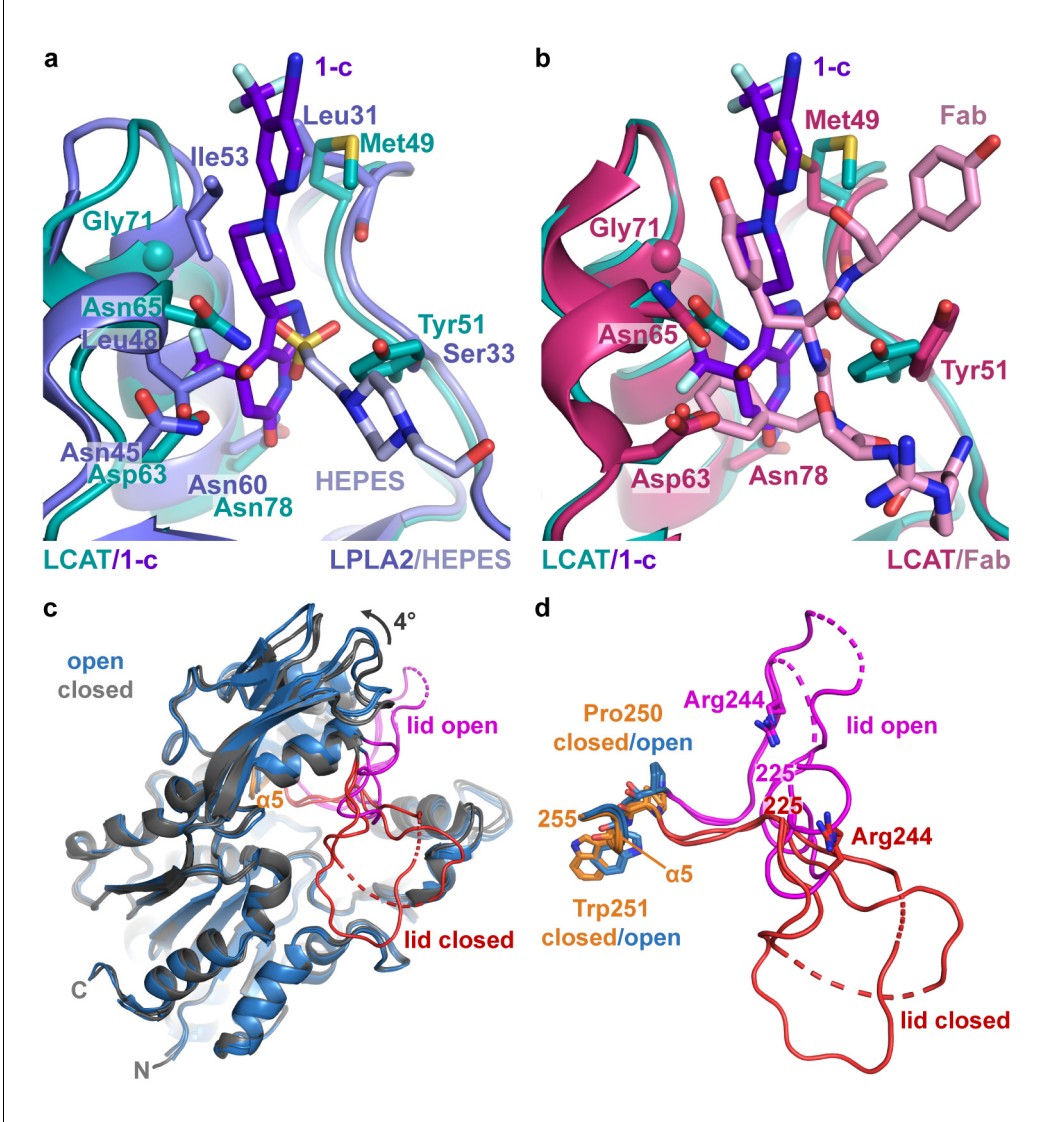

**Figure 3.** Comparison of LCAT and LPLA2 structures. (a) ΔNΔC-IDFP·1 structure aligned with LPLA2 (blue, PDB entry 4X90) bound to HEPES (light blue). Residues that are not conserved within the binding pocket are labeled and shown as stick models. (b) Structure of ΔNΔC-IDFP·1 aligned to that of 27C3–LCAT–Fab1 (dark pink, PDB entry 5BV7 with Fab1 shown in pink), highlighting residues that adjust conformation to accommodate the different ligands. (c) Four LCAT crystal structures aligned to show differences between the open and closed states. Closed (presumably inactive) structures are shown in gray (PDB entries 4XWG and 5TXF) with orange hinge and red lid. Open structures (structure reported here and 27C3–LCAT–Fab1) are shown in blue with magenta lid. Dashed lines indicate disordered residues. (d) Close up of structures from (c) only depicting the lid and hinge region. Hinge residues Pro250 and Trp251 and lid residue Arg244 are shown as stick models.

DOI: https://doi.org/10.7554/eLife.41604.013

The following figure supplements are available for figure 3:

**Figure supplement 1.** The ΔNΔC-IDFP·1 structure has lower temperature factors in the membrane binding domain and lid.

DOI: https://doi.org/10.7554/eLife.41604.014

**Figure supplement 2.** Hinge and lid movement modulate lipid binding tracks.

DOI: https://doi.org/10.7554/eLife.41604.015

**Figure supplement 3.** Structure-activity relationships.

DOI: https://doi.org/10.7554/eLife.41604.016

hydrophobic track corresponding to track A that takes a straighter path to the back of the LCAT as compared the one observed for LPLA2, which results from the different orientations of their lids (*Figure 2—figure supplement 3b–c*). We previously used HDX MS to show that IDFP stabilizes elements in the MBD and the lid region of LCAT (*Manthei et al., 2017*). This data is in agreement with what we observe in the crystal structure of ΔNΔC-IDFP·**1**, in that residues 67–72 in the MBD and residues 226–236 in the lid have markedly lower temperature factors in the structure reported here as compared to LCAT structures without IDFP (*Figure 3—figure supplement 1*).

## Comparison with prior LCAT structures reveals a global conformational switch

Reported atomic structures of LCAT include that of full-length LCAT wherein its lid extends over and shields the active site (PDB entry 5TXF, LCAT-closed), LCAT in complex with inhibitory Fab1 (LCAT–Fab1), and LCAT in complex with Fab1 and a second agonistic Fab fragment (27C3) (entry 5BV7, 27C3–LCAT–Fab1; *Figure 3c*). In these structures, the N- and C-termini are disordered except for an N-terminal pentapeptide in the 27C3–LCAT–Fab1 structure (containing mutations L4F/N5D) that docks in the active site of a neighboring symmetry mate. It is unclear which of these structures, if any, represent an activated conformation of LCAT, although the LCAT–Fab1 and LCAT-closed structures are more similar to each other and likely to be inactive, whereas 27C3–LCAT–Fab1 has a more exposed active site. The conformation of the active site lid is highly variable among these three structures.

The ΔNΔC-IDFP·**1** structure affords a high-resolution view of LCAT in what is expected to be a fully activated conformation unobstructed by conformational changes that might be induced by Fab binding. The structure of LCAT here is most similar to that in 27C3–LCAT–Fab1 (RMSD 0.70 Å for all Cα atoms) (*Gunawardane et al., 2016*; *Krissinel and Henrick, 2004*), including in their active site lid regions and in the relative configuration of their three domains (*Figure 3—figure supplement 1a–b*). The active site lid can be divided into two regions, with the C-terminal portion (residues 233–249) being most consistent between the two structures. Both structures contain similar disordered segments (residues 236–242 in 27C3–LCAT–Fab1, chain A residues 239–240 and chain B residues 236–242 in ΔNΔC-IDFP·**1**). The N-terminal portion of the lid (residues 225–232) is most variable, although it is consistent between the two unique chains of the ΔNΔC-IDFP·**1** structure and, given the substrate analog, more likely to adopt a physiological conformation. Indeed, the N-terminal pentapeptide of a symmetry mate in the 27C3–LCAT–Fab1 structure would clash with Asn228 in the lid region of ΔNΔC-IDFP·**1**. Regardless, such differences highlight the high plasticity of the active site, which is likely required for LCAT to accommodate its various lipidic substrates.

Comparison of the structure of LCAT-closed with ΔNΔC-IDFP·**1** provides a unique glimpse of how LCAT transitions from inactive to active states (*Video 1*). Domain motion analysis (*Hayward and Berendsen, 1998*) reveals two hinge regions: residues 219–229 and 251–255 (*Video 2*). The dihedral angles between Asn228-Gln229 and Gln229-Gly230 undergo a large rotation that flips the lid region away from the active site in the ΔNΔC-IDFP·**1** complex. On the other end of the lid, the α5 helix of the cap

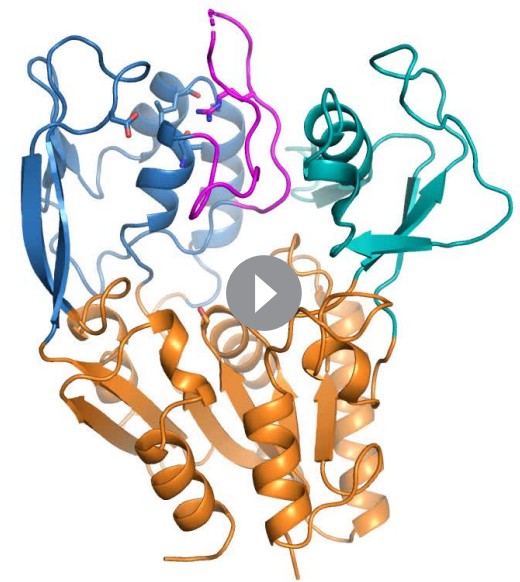

**Video 1.** Transition between closed and open conformations of LCAT. The video highlights the opening of the lid and corresponding cap domain movements that occur upon LCAT activation. Arg244 and the residues it interacts with in each conformation, as well as the active site location Ser181 are shown as sticks. Chimera (*Pettersen et al., 2004*) was used to morph from the closed structure (PDB entry 5TXF (*Manthei et al., 2017*)) to the activator structure. The video was rendered using PyMOL.
DOI: https://doi.org/10.7554/eLife.41604.017

domain unwinds in the lid open state, with the dihedral angles between Pro250-Trp251 undergoing the most change (*Figure 3c–d*, *Video 2*). The lid transition is accompanied by a 4° change in the orientation of the adjacent cap domain relative to both the α/β-hydrolase and MBD, which remain fixed with respect to each other (*Figure 3c*, *Video 1*). Interestingly, in all reported LCAT structures the binding site for compound **1** is accessible (with obvious exception of those in complex with Fab1, which takes advantage of the same site), regardless of the orientation of the cap domain. In other words, initial HDL-binding and subsequent occupation of the active site by a ligand are most likely responsible for triggering the lid opening and rearrangement of the cap domain we observe in the structure, and not the binding of **1**.

As a consequence of these conformational changes in the lid and reorientation of the cap domain, there are alterations within the active site that likely facilitate binding to substrates. In LPLA2, two distinct tracks for the acyl chains of lipid substrates were observed (*Figure 2—figure supplement 3c*) (*Glukhova et al., 2015*). Track A is furthest from the lid loop and is only solvent-accessible when the lid is retracted, and the α5 helix, including hinge residue Trp251, unwinds and moves inwards to block this track in the closed lid conformation of LCAT-closed (*Figure 3—figure supplement 2*). In the lid-open structures, Lys218 moves with the cap domain away from the MBD in the activated conformation, where it would be in better position to bind the phosphate in the substrate lipid head group (*Glukhova et al., 2015*) (*Figure 3—figure supplement 2a*).

## Structure-activity relationships

The structure of the ΔNΔC-IDFP·**1** complex confirms the structure-activity relationships we and others have observed for the pyrazolopyridine scaffold. The hydrogen bonds formed by the pyrazole ring with the backbone carbonyl of Met49 and amide of Tyr51 (*Figure 2b–c*) indicate that **1-b** is the dominant tautomerized isoform in the co-crystallized structure (*Figure 2—figure supplement 2a*). Although the exchange of pyrazole (**2**) to imidazole (**3**) eliminates the hydrogen bond with Met49, this resulted in only a minimal change in $EC_{50}$ (280 and 320 nM for **2** and **3**, respectively) and no change in the maximum response (*Tables 1* and *2*). However, interruption of both of these hydrogen bonds by swapping the pyrazole (**2**) for isoxazole (**9**, *Figure 2—figure supplement 2b*) dramatically increased the $EC_{50}$ to 7.7 μM and decreased the response to 1.6-fold (*Tables 2* and *5*). It was previously shown that removal of the C4 hydroxyl group (**4**, *Figure 2—figure supplement 2b*), which interacts with Asp63 in the structure, caused a ~ 6 fold drop in potency compared with **2** (*Kobayashi et al., 2015a*; *Kobayashi et al., 2015b*). This is consistent with elimination of the hydroxyl group of **3** to give the more planar structure of **8** (*Figure 2—figure supplement 2b*) which decreased the potency to 4.6 μM, yet interestingly it still activated LCAT with increased efficacy of 3.7-fold (*Tables 2* and *5*). Surprisingly, although the bicyclic head of these compounds is expected to play an important role in the retention of potency, the imidazole-containing head group of **3** has no activating effect at concentrations up to 10 μM (**6**, *Figure 2—figure supplement 2b*), perhaps due to loss of favorable interactions with Met49. Consistent with the above data, compounds **6**, **8**, and **9** could not thermal stabilize LCAT at 10 μM in DSF, although **8** could at 100 μM (*Figure 3—figure supplement 3a*). MST further confirmed that compound **6** was unable to bind to LCAT (*Figure 3—figure supplement 3b*). Thus, in this series of activators, potency and efficacy are therefore highly dependent on a hydroxyl and chirality at the C4 position, as well as maintenance of a pyrazine ring system that likely assists in interactions with hydrophobic substrates.

## Perturbation of the activator binding site

To further validate the crystal structure and better understand the mechanistic role of the MBD, we exchanged residues in the activator binding site of LCAT with their equivalents in LPLA2, which is not stabilized by **1** or related compounds (*Figure 1g*). The Y51S, G71I, and Y51S/G71I (*Figure 3a*) variants were thus expected to be impaired in binding. These variants exhibited similar or higher $T_m$ values than WT LCAT, and were able to hydrolyze both the soluble substrate *p*-nitrophenyl butyrate (pNPB) and the micellar substrate MUP (*Figure 4*, *Figure 4—figure supplement 1*), indicating an intact fold. As expected, compound **1** was far less effective at increasing the $T_m$ of the three variants compared to WT (*Figure 4a*). The Y51S/G71I variant also exhibited a nearly 4-fold decrease in HDL binding affinity and a reduced ability to catalyze acyl transfer (*Figure 4b–c*, *Figure 4—figure supplement 2*). These results are consistent with recent studies probing nearby positions at Trp48 (mutated

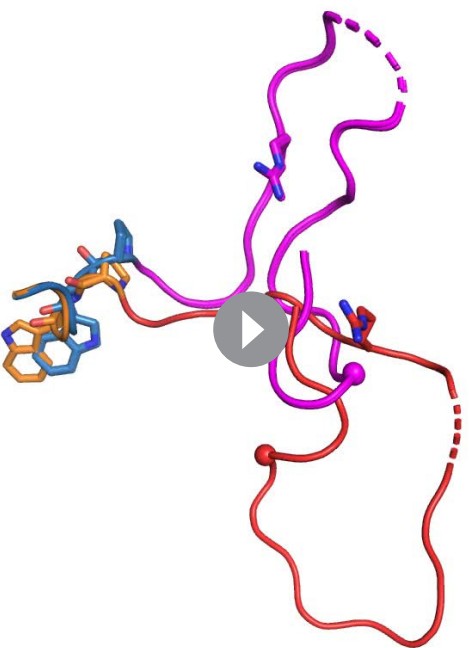

**Video 2.** Movement corresponding to the hinge region. The same morph as depicted in *Video 1*, but zoomed in on the lid and hinge region. The closed (presumably inactive) structure (PDB entry 5TXF (*Manthei et al., 2017*)) is shown with orange hinge and red lid. The ΔNΔC-IDFP·**1** structure is shown in blue with magenta lid, which is retained during the morph. Dashed lines indicate disordered residues. Hinge residues Pro250 and Trp251 are shown as stick models, as well as the side chain of Arg244 in the lid region. The position of the Cα atom of Gly230 is indicated with a sphere.
DOI: https://doi.org/10.7554/eLife.41604.018

to Ala) and Leu70 (mutated to Ser) (*Manthei et al., 2017*) or the analogous positions in LPLA2 (*Glukhova et al., 2015*). Conversely, the analogous LPLA2 chimeric variants (S33Y, I53G, and S33Y/I53G) had lower $T_m$ values relative to WT (*Figure 4a*). However, these variants remained unable to be stabilized by **1**. We were unable to express and test a triple mutant expected to fully restore binding (S33Y/I53G/L48N).

Compound **1** did not stimulate pNPB esterase activity for any variant of LCAT (*Figure 4—figure supplement 1b*), and in fact seemed to inhibit the activity of WT. Perturbation of the activator binding site decreased this effect. Compound **1** and related compounds activated hydrolysis in the MUP assay (*Figure 4d*, *Tables 1* and *2*). The EC$_{50}$ of Y51S with **1** was 4-fold higher than WT at 0.59 μM, G71I had an EC$_{50}$ >5 μM, and Y51S/G71I had no response at concentrations up to 10 μM **1**. We confirmed these results in a DHE acyltransferase assay with the Y51S/G71I variant, wherein the mutation failed to increase activity in the presence of compound **2** (*Figure 4c,e*, *Table 1*). These results confirm that the binding site for compound **1** in the crystal structure is responsible for the biochemical effects observed in solution.

## Recovery of activity in an FLD variant

Arg244 is a position commonly mutated in LCAT genetic disease (R244G (*McLean, 1992*; *Vrabec et al., 1988*), R244H (*Pisciotta et al., 2005*; *Sampaio et al., 2017*; *Strøm et al., 2011*), R244C (*Charlton-Menys et al., 2007*), and R244L (*Castro-Ferreira et al., 2018*)) and its side chain forms unique interactions in the observed active and inactive states of LCAT. In data obtained from patient plasma, the amount of LCAT-R244G isolated from homozygotes was ~25% of the amount from WT LCAT plasma and there was ~15% of WT LCAT activity, whereas heterozygotes of the

**Table 5.** EC$_{50}$ values of LCAT variants in the MUP esterase assay with compounds **6**, **8**, and **9**.

| | EC$_{50}$ (μM) | | |
|---|---|---|---|
| Variant\Compound | 6 | 8 | 9 |
| WT | no effect | 4.6 ± 0.06 | 7.7 ± 2 |
| Y51S | no effect | >10 | >10 |
| G71I | no effect | >10 | no effect |
| Y51S/G71I | no effect | >10 | no effect |
| R244A | no effect | >10 | 6.2 ± 0.8 |
| R244H | no effect | >10 | 7.6 ± 1 |

Compounds were titrated from 0.04 to 9.5 μM, and reactions were performed in triplicate with values reported as mean ± s.e.m.
DOI: https://doi.org/10.7554/eLife.41604.019

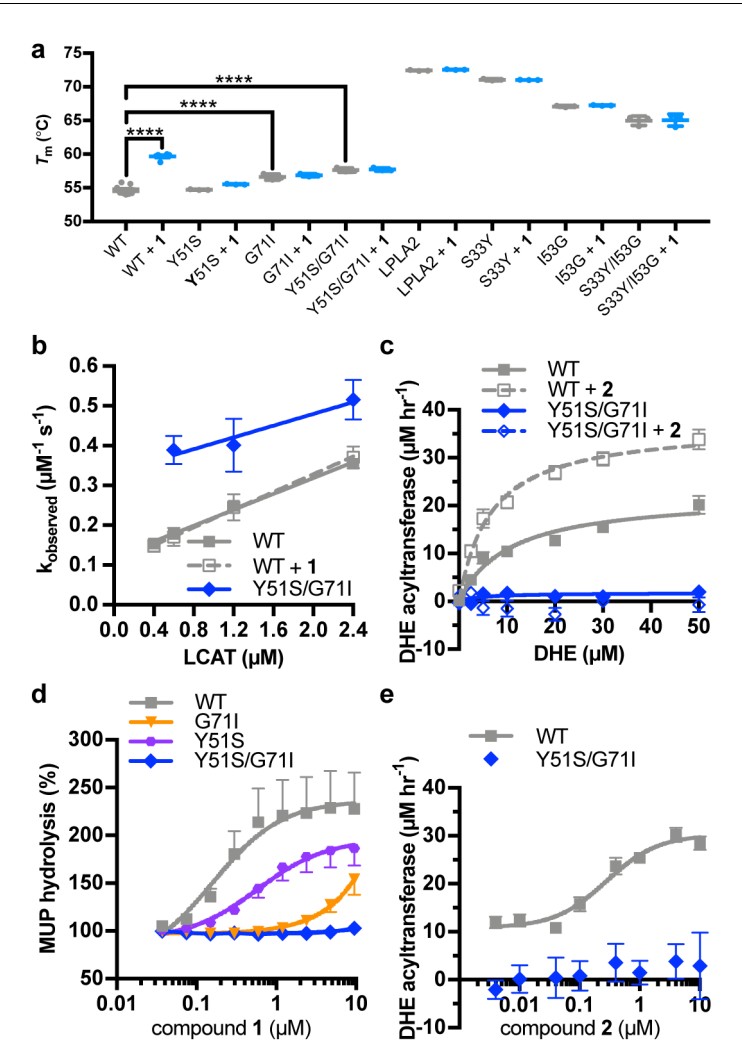

**Figure 4.** Characterization of activator binding site mutants. (a) Perturbation of the activator binding site leads to loss of responsiveness to compound **1**, although the G71I and Y51S/G71I variants are themselves stabilized compared to WT LCAT. LPLA2 variants, however, do not bind to **1**, and chimeric swaps are destabilized. Data are mean ± s.e.m. of at least three independent experiments performed in duplicate. ****p<0.0001 by one-way analysis of variance followed by Tukey's multiple comparisons post-test. Each protein without ligand was compared to that same variant with compound **1**, and WT LCAT was compared to each LCAT variant. Non-significant comparisons are not shown. (b) Plot used to determine $k_{on}$, $k_{off}$, and hence $K_d$ from BLI data for LCAT binding to HDL. Data are mean ± s.e.m. of three independent experiments. (c) DHE acyltransferase assay with peptide HDLs comparing the absence (solid lines) and presence (dashed lines) of 5 µM compound **2**, which was used in this assay instead of **1** due to its lower background fluorescence. Data are mean ± s.e.m. of three independent experiments performed in triplicate. (d) Titration of compound **1** in the MUP hydrolysis assay. Data were normalized to basal activity of 100% for each variant to give percent activation. Data are mean ± s.e.m. of three independent experiments. (e) Titration of compound **2** in the DHE acyltransferase assay. Data are mean ± s.e.m. of three independent experiments performed in triplicate.

DOI: https://doi.org/10.7554/eLife.41604.020

The following figure supplements are available for figure 4:

**Figure supplement 1.** Biochemical characterization of LCAT variants.
DOI: https://doi.org/10.7554/eLife.41604.021
**Figure supplement 2.** Representative BLI data for LCAT-Arg244 variants.
DOI: https://doi.org/10.7554/eLife.41604.022

R244G and R244H mutations had ~80% and~50% of WT LCAT activity, respectively (*Pisciotta et al., 2005*; *Vrabec et al., 1988*), thus supporting an important role for this residue. Arg244 is found in the lid of LCAT and interacts with the backbone carbonyls of Leu223 and Leu285 in ΔNΔC-IDFP·**1**, and with the side chain of Asp335 in the lid closed state of LCAT-closed (*Figure 5a*, *Video 1*). We hypothesized that molecules targeting the MBD could restore some stability and function of mutations at Arg244 because this residue does not participate in the binding site for **1**. The LCAT-R244A and -R244H variants were purified and shown to be less stable than WT with $\Delta T_m$ values of −2.3 and −2.4 °C, respectively, consistent with Arg244 playing an important structural role (*Figure 5b*, *Figure 4—figure supplement 1a*). Both LCAT-R244A and -R244H exhibited WT levels of pNPB activity, but 44% and 78% of WT in the MUP hydrolysis assay (*Figure 4—figure supplement 1b–c*). In HDL binding analyses, both variants had an increased $k_{off}$ (2-fold for R244A and 3.5-fold for R244H) which led to an increase in their overall $K_d$ values (*Figure 4—figure supplement 2*, *Table 3*). For R244H, the $k_{on}$ was also decreased from 0.091 (WT) to 0.022 $\mu M^{-1} s^{-1}$. Thus, in the context of HDL binding, the histidine mutant is less tolerated, perhaps due to steric clashes in the lid open conformation. Neither variant had substantial activity in the acyltransferase assay (*Figure 5c*), consistent with their contribution to FLD.

R244A and R244H were both stabilized by the addition of compound **1** ($\Delta T_m$ of 6.0°C and 4.8 °C, respectively, *Figure 5b*). R244A, R244H, and WT LCAT all exhibited similar $EC_{50}$ values in response to **1** in the MUP esterase assay (~150 nM), with all three variants being activated about 2-fold by compounds **1**–**3** (*Figure 5d*, *Tables 1* and *2*). In the DHE acyltransferase assay, the $EC_{50}$ values in the presence of saturating compound **2** were 0.28, 0.76, and ≥4.6 $\mu M$ for WT, R244A, and R244H,

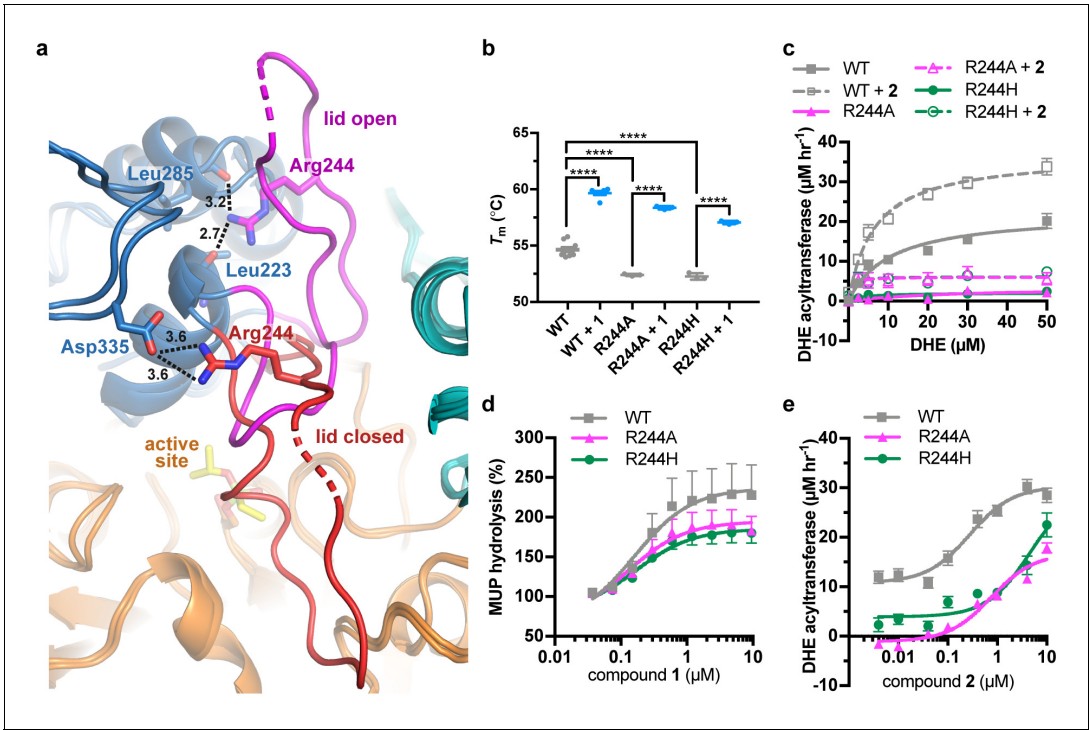

**Figure 5.** LCAT-Arg244 variants can be partially rescued by LCAT activators. (**a**) LCAT-Arg244 acts as part of a molecular switch that interacts with the backbone carbonyls of Leu223 and Leu285 in activated structures of LCAT (magenta lid). In an inactive structure (red lid, PDB entry 5TXF), Arg244 instead interacts with the side chain of Asp335. Hydrogen bonds are indicated by black dashed lines with distances in Å. (**b**) The Arg244 variants have lower $T_m$ values relative to WT, yet compound **1** can stabilize each to the same extent. Data are mean ± s.e.m. of at least three independent experiments performed in duplicate. ****p<0.0001 by one-way analysis of variance followed by Tukey's multiple comparisons post-test. (**d**) DHE acyltransferase assay with peptide-based HDLs comparing the absence (solid lines) and presence (dashed lines) of 5 $\mu M$ **2**. Data are mean ± s.e.m. of three independent experiments performed in triplicate. (**c**) Titration of **1** in the MUP esterase assay. Data were normalized to basal activity of 100% for each variant to give percent activation. Data are mean ± s.e.m. of three independent experiments. (**d**) Titration of compound **2** in the DHE acyltransferase assay. Data are mean ± s.e.m. of three independent experiments performed in triplicate.
DOI: https://doi.org/10.7554/eLife.41604.023

respectively (*Figure 5e*, *Table 1*). At the highest concentration tested (10 μM compound **2**), the acyl-transferase rate was 18 and 26 μM h$^{-1}$ for R244A and R244H, respectively, both greater than WT LCAT which had a rate of 11 μM h$^{-1}$ at the lowest concentration of compoud **2** examined. The activator affected HDL binding of the two Arg244 variants differently. For R244A, compound **1** decreased the $k_{on}$ from 0.069 to 0.017 μM$^{-1}$ s$^{-1}$, which increases the $K_d$ from 3.2 to 11 μM. For R244H, compound **1** enhanced binding to HDL by reducing the $k_{off}$ from 0.40 to 0.15 s$^{-1}$, reducing the $K_d$ from 18 to 4.3 μM (*Figure 4—figure supplement 2*, *Table 3*). Thus, piperidinylpyrazolopyridine and piperidinylimidazopyridine activators like compound **1** can partially rescue defects in the activity of LCAT-Arg244 variants.

## Discussion

Here we have defined a novel activator binding site in the MBD of LCAT as well as the active conformation of LCAT, and have demonstrated that these activators can restore the activity of some FLD variants. However, the mechanism of activation mediated by compound **1** and its analogs is not straightforward. The activators do not alter the binding constant of WT LCAT for HDL (*Figure 1e*, *Table 3*), suggesting that they do not contribute to HDL binding despite occupying a site in the MBD. Thus, one would expect that the residues that interact with compound **1** are not involved in HDL binding, or else these compounds would act as inhibitors. However, the site is closely juxtaposed with residues that are involved in HDL binding. HDL-binding residues such as Trp48 and Leu70 are adjacent to the activator binding site (*Manthei et al., 2017*), and the double mutant Y51S/G71I was 4-fold decreased in its affinity for HDLs due to a defect in the $k_{off}$, and lost acyltransferase activity (*Figure 4*, *Table 3*). A G71R variant has also been reported in LCAT genetic disease (*Hörl et al., 2006*).

The compounds increase activity of WT LCAT up to 3.7-fold, specifically by increasing the $V_{max}$, although by acting at a site remote from the catalytic triad and IDFP binding site (*Figure 1*, *Table 2*). The typical mechanism for acting at a distance would be allostery, wherein ligand binding induces a conformational change that alters the active site. Indeed, ΔNΔC-IDFP·**1** adopts what we believe is a more active state with alterations in the active site that should promote activity. However, the MBD of LCAT does not appreciably change its orientation with respect to the hydrolase domain in any reported structure thus far, and the activator binding site seems available regardless of LCAT conformation. Moreover, the increase in $T_m$ caused by IDFP and compound **1** is additive, not synergistic (*Figure 1f–g*), and our previous HDX MS data suggested that IDFP alone can stabilize LCAT in an active, lid open conformation that is likely represented by the current structure (*Manthei et al., 2017*). Thus, IDFP is more likely to be the driver of the observed global conformation change observed in the crystal structure of ΔNΔC-IDFP·**1**. Although both ligands stabilize, they do so via independent mechanisms and compound **1** may only do so locally.

Thus, we hypothesize that the activators such as compund **1** act by stabilizing the MBD and facilitating substrate entry into the active site cleft of the enzyme (*Figure 6*). In support of such a model, we note that the two chains of LCAT in the asymmetric unit of the ΔNΔC-IDFP·**1** crystals pack to form a pseudo-symmetric homodimer utilizing an interface with many of the hydrophobic residues from the MBD including Trp48, Leu64, Phe67, Leu68, Pro69, Leu70 and Leu117 from the αA-αA′ loop (*Figure 2—figure supplement 3a*, *Figure 6—figure supplement 1*). The interface is centered on the side chains of Leu64 and Phe67. The pyrazine ring of the activator is prominently featured in this hydrophobic surface. This hydrophobic ring packs next to residues in the MBD well-known to be important for membrane interactions, such as the conspicuously solvent exposed Trp48 side chain (*Figure 6—figure supplement 1a*). This same interface was also proposed by a recent molecular dynamics study exploring the ability of LCAT to dock to a model membrane in both the closed and open conformations (*Casteleijn et al., 2018*). In the closed conformation, the active site lid blocks Leu64, Phe67, and Leu117 from being able to access membranes, though the rest of the MBD and the hydrophobic N-terminus of LCAT, which is also key for HDL binding (*Manthei et al., 2017*), would still be available (*Figure 6—figure supplement 1b*). The simulations in this study also suggested that residues such as Phe67 were involved in promoting transfer of lipids into the active site tunnel of the enzyme. Mutation of Arg244, unlike compound **1**, clearly affects binding to HDL, and thus this residue, or the lid region in which it resides, could be a major ApoA-I binding determinant

**Figure 6.** Model of LCAT activation. We previously described the closed lid structure of LCAT (middle) wherein the lid (magenta coil) would shield the hydrophobic active site of LCAT in solution. In the crystal structure described herein, LCAT adopts an open conformation (left), which is stabilized by IDFP bound in the active site (yellow star) and the small molecule activator (purple ellipse). We hypothesize that LCAT binds to discoid HDL with a similar open lid conformation and that the activator facilitates lipid transport into the active site. Thus the hydrophobic interface is shown binding to the hydrophobic acyl chains on the side of discoidal HDL, and the expected ApoA-I (green helices) interaction is shown as a contact with the lid, providing a structural explanation for ApoA-I activation of LCAT (*Cooke et al., 2018*). LCAT is depicted with the α/β-hydrolase domain in orange, the cap domain in blue, and the membrane binding domain in teal. The R244 side chain in the lid is shown as pink sticks. The dashed orange line in the HDL complex depicts the disordered N-terminus of LCAT which is also critical for HDL binding.

DOI: https://doi.org/10.7554/eLife.41604.024

The following figure supplement is available for figure 6:

**Figure supplement 1.** The activator molecule contributes to a hydrophobic surface.

DOI: https://doi.org/10.7554/eLife.41604.025

(*Figure 4—figure supplement 2*, *Table 3*). Indeed, a recent paper identified a crosslink between LCAT and ApoA-I at nearby residue Lys240 within the lid (*Cooke et al., 2018*).

A better understanding of how ligands fit within the activator pocket enables rational design to create more potent and effective LCAT activators. For example, our crystal structure revealed the preferred enantiomer of bound piperidinylpyrazolopyridines, thus one could expect at least two-fold higher potency could be achieved with an enantiopure preparation. A recent patent has improved the potency of these compounds 3-fold by using an optically pure compound, as well as adding a hydroxyl to the C5 position on the bicyclic head, which our structure indicates would add a second hydrogen bond with the side chain of Asp63 (*Kobayashi et al., 2016*). Furthermore, we have shown that there is potential to increase the efficacy of the compounds, because compound **8** activated 3.7-fold compared to the parent compounds, which activated an average of 2.3-fold. However, **8** had lowerpotency, and so more modulations will be required to determine if potency and efficacy can be improved simultaneously.

The ability to perform rational design is important because we also demonstrated here the therapeutic potential of using small molecule activators targeting the MBD in FLD patients. We focused on mutations at Arg244 (*Castro-Ferreira et al., 2018*; *Charlton-Menys et al., 2007*; *McLean, 1992*; *Pisciotta et al., 2005*; *Sampaio et al., 2017*; *Strøm et al., 2011*; *Vrabec et al., 1988*) because of its apparent role in the switch mechanism of the active site lid, but in principle any patient harboring an alternative missense mutation that does not directly perturb the hydrolase active site may also benefit from this compound series. In this sense the ability of piperidinylpyrazolopyridine LCAT activators to rescue Arg244 mutants parallels the allosteric action of ivacaftor on the G551D mutant of the cystic fibrosis transmembrane conductance regulator, although their mechanisms of action are necessarily different due to the unique structure of the MBD (*McPhail and Clancy, 2013*). Even a relatively small increase in activity could potentially slow or reverse the progression of renal disease in some FLD patients because FED patients with only partial LCAT activity do not develop renal disease (*Ahsan et al., 2014*). Certainly, treatment with a small molecule activator would be more cost effective and easier for patients comply with than rhLCAT enzyme replacement therapy. In future experiments, it will be important to examine the utility of activators like compound **1** for other FLD variants. Lastly, because these compounds were demonstrated to effectively increase HDL-C in

monkeys with normal levels of LCAT (*Kobayashi et al., 2016*; *Kobayashi et al., 2015a*; *Kobayashi et al., 2015b*; *Onoda et al., 2015*), it will be important to continue to interrogate their mechanism and determine if they also increase cholesterol efflux and promote atherosclerotic plaque regression. If so, then activation of LCAT by a small molecule approach and improving HDL function could be widely used in the primary prevention of cardiovascular disease and would likely complement our existing drugs for lowering LDL-C, such as statins and PCSK9-inhibitors.

# Materials and methods

**Key resources table**

| Reagent type (species) or resource | Designation | Source or reference | Identifiers | Additional information |
|---|---|---|---|---|
| Cell line (*Homo sapiens*) | HEK293F | ThermoFisher | ThermoFisher: R79007 | |
| Cell line (*H. sapiens*) | HEK293F ΔNΔC-LCAT | this paper | | polyclonal stable cell line |
| Recombinant DNA reagent | pcDNA4 LCAT | (*Glukhova et al., 2015*) DOI: 10.1038/ncomms7250 | | |
| Recombinant DNA reagent | pcDNA4 ΔNΔC-LCAT | (*Glukhova et al., 2015*) DOI: 10.1038/ncomms7250 | | |
| Recombinant DNA reagent | pProEX HT-EndoH | other | | D. J. Leahy, Johns Hopkins |
| Peptide, recombinant protein | ESP24218 peptide | Genscript | | PVLDLFRELLNE LLEALKQKLK |
| Commercial assay or kit | Index HT screen | Hampton Research | Hampton: HR2-134 | |
| Commercial assay or kit | Monolith His-Tag Labeling Kit RED-tris-NTA 2nd Generation | Nanotemper Technologies | Nanotemper: MO-L018 | |
| Chemical compound, drug | 1,2-dipalmitoyl-sn-glycero-3-phosphocholine (DPPC) | Avanti Polar Lipids | Avanti: 850355 | |
| Chemical compound, drug | 1-palmitoyl-2-oleoyl-sn glycero-3-phosphocholine (POPC) | NOF America | NOF America: MC-6081 | |
| Chemical compound, drug | 16:0 Biotinyl Cap PE | Avanti Polar Lipids | Avanti: 870277 | |
| Chemical compound, drug | 4-Methylumbelliferyl phosphate | Cayman Chemical | Cayman: 16089 | |
| Chemical compound, drug | Bio-beads SM-2 | Bio-Rad | Bio-Rad: 1523920 | |
| Chemical compound, drug | Bovine Serum Albumin (BSA), fatty acid free | Sigma-Aldrich | Sigma: A8806 | |
| Chemical compound, drug | Cholesterol oxidase | Sigma-Aldrich | Sigma: C8649 | |
| Chemical compound, drug | Dehydroergosterol (DHE) | Sigma-Aldrich | Sigma: E2634 | |
| Chemical compound, drug | DMEM high glucose with GlutaMAX and 1 mM pyruvate | ThermoFisher Scientific | ThermoFisher: 10569 | |
| Chemical compound, drug | Fetal bovine serum | Sigma-Aldrich | Sigma: F2442 | |

*Continued on next page*

*Continued*

| Reagent type (species) or resource | Designation | Source or reference | Identifiers | Additional information |
|---|---|---|---|---|
| Chemical compound, drug | FreeStyle 293 Expression Medium | ThermoFisher Scientific | ThermoFisher: 12338026 | |
| Chemical compound, drug | Isopropyl dodecyl fluorophosphonate | Cayman Chemical | Cayman: 10215 | |
| Chemical compound, drug | Kifunensine | Cayman Chemical | Cayman: 10009437 | |
| Chemical compound, drug | Ni-NTA | Qiagen | Qiagen: 30230 | |
| Chemical compound, drug | OptiMEM Reduced Serum Medium | ThermoFisher Scientific | ThermoFisher: 31985070 | |
| Chemical compound, drug | Penicillin-Streptomycin | ThermoFisher Scientific | ThermoFisher: 15140122 | |
| Chemical compound, drug | Polyethylenimine (PEI) | Polysciences | Polysciences: 23966 | |
| Chemical compound, drug | p-nitrophenyl butyrate (pNPB) | Sigma-Aldrich | Sigma: N9876 | |
| Chemical compound, drug | SspI | New England Biolabs | New England Biolabs: R0132S | |
| Chemical compound, drug | SYPRO Orange | ThermoFisher Scientific | ThermoFisher: S6650 | |
| Chemical compound, drug | 6-(4-(4-Hydroxy-6-oxo-4-(trifluoromethyl)—4,5,6,7-tetrahydro-1H-pyrazolo[3,4-b]pyridin-3-yl)piperidin-1-yl)—4-(trifluoromethyl)nicotinonitrile, TFA (compound 1) | example 95 in US patent 9150575 | | |
| Chemical compound, drug | 4-Hydroxy-4-(trifluoromethyl)—3-(1-(5-(trifluoromethyl)pyrazin-2-yl)piperidin-4-yl)—4,5-dihydro-1H-pyrazolo[3,4-b]pyridin-6(7H)-one (compound 2) | example 46 in US patent 9150575 | | |
| Chemical compound, drug | 7-Hydroxy-7-(trifluoromethyl)—1-(1-(5-(trifluoromethyl)pyrazin-2-yl)piperidin-4-yl)—6,7-dihydro-1H-imidazo[4,5-b]pyridin-5(4H)-one (compound 3) | example 3 in WO patent 2015087996A1 | | |
| Chemical compound, drug | 4-(trifluoromethyl)—3-(1-(5-(trifluoromethyl)pyrazin-2-yl)piperidin-4-yl)—1,4,5,7-tetrahydro-6H-pyrazolo[3,4-b]pyridin-6-one (compound 4) | example 10 in WO patent 2015111545A1 | | |
| Chemical compound, drug | 4-Hydroxy-3-(piperidin-4-yl)—4-(trifluoromethyl)—4,5-dihydro-1H-pyrazolo[3,4-b]pyridin-6(7H)-one, HCl (compound 5) | reference example 60 in US patent 9150575 | | |
| Chemical compound, drug | 7-Hydroxy-1-(piperidin-4-yl)—7-(trifluoromethyl)—6,7-dihydro-1H-imidazo[4,5-b]pyridin-5(4H)-one, HCl (compound 6) | reference example 1 in WO patent 2015087996A1 | | |
| Chemical compound, drug | 4-Hydroxy-3-(piperidin-4-yl)—4-(trifluoromethyl)—4,5-dihydroisoxazolo[5,4-b]pyridin-6(7H)-one, HCl (compound 7) | this paper | | Compound was synthesized by NCATS |
| Chemical compound, drug | 7-(trifluoromethyl)—1-(1-(5-(trifluoromethyl)pyrazin-2-yl)piperidin-4-yl)—1H-imidazo[4,5-b]pyridin-5(4H)-one (compound 8) | this paper | | Compound was synthesized by NCATS |

*Continued on next page*

*Continued*

| Reagent type (species) or resource | Designation | Source or reference | Identifiers | Additional information |
|---|---|---|---|---|
| Chemical compound, drug | 4-hydroxy-4-(trifluoromethyl)—3-(1-(5-(trifluoromethyl)pyrazin-2-yl)piperidin-4-yl)—4,5-dihydroisoxazolo[5,4-b]pyridin-6(7H)-one (compound 9) | this paper | | Compound was synthesized by NCATS |
| Software, algorithm | HKL-2000 | (*Otwinowski and Minor, 1997*) DOI: 10.1016/S0076-6879(97)76066-X | | |
| Software, algorithm | PHASER | (*McCoy, 2007*) DOI: 10.1107/S0907444906045975 | | |
| Software, algorithm | REFMAC5 | (*Murshudov et al., 2011*) DOI: 10.1107/S0907444911001314 | | |
| Software, algorithm | Phenix | (*Adams et al., 2010*) DOI: 10.1107/S0907444909052925 | | |
| Software, algorithm | Coot | (*Emsley et al., 2010*) DOI: 10.1107/S0907444910007493 | | |
| Software, algorithm | Molprobity | (*Chen et al., 2010*) DOI: 10.1107/S0907444909042073 | | |
| Software, algorithm | Prism 7.0 c | Graphpad Software | | |
| Software, algorithm | Octet Data Analysis 7.0 | FortéBio | | |
| Software, algorithm | PyMOL Molecular Graphics System | Schrödinger | | |
| Software, algorithm | Chimera | (*Pettersen et al., 2004*) DOI: 10.1002/jcc.20084 | | |
| Software, algorithm | Protein Thermal Shift | ThermoFisher Scientific | | |
| Software, algorithm | MO.Affinity Analysis | Nanotemper Technologies | | |

## Cell culture, Protein Production, and Purification

To produce protein for crystallographic screens, a stable cell line expressing ΔNΔC-LCAT was created in HEK293F cells. A codon-optimized human ΔNΔC-LCAT construct with a C-terminal 6x histidine-tag in pcDNA4 was SspI digested and transfected into HEK293F cells. Cells were selected with zeocin and grown in adherent culture on 150 mm plates in Dulbecco's Modified Eagle Medium high glucose medium with GlutaMAX and 1 mM pyruvate, supplemented with 10% fetal bovine serum (Sigma), 100 U/ml penicillin, 100 µg/ml streptomycin and 50 µg mL$^{-1}$ zeocin. Kifunensine was added to 5 µM once the cells were confluent to prevent complex glycosylation. Conditioned media was harvested every 5 days, purified via Ni-NTA, dialyzed against reaction buffer (20 mM HEPES pH 7.5, 150 mM NaCl), and then frozen. For crystallographic trials, samples were thawed and subsequently cleaved with a 1:3 endoglycosidase H:LCAT molar ratio in reaction buffer supplemented with 100 mM NaOAc pH 5.2 for 2.5 hr at room temperature, which reduces the heterogeneous *N*-glycans to single *N*-acetylglucosamines. HEPES pH 8 was then added to 100 mM prior to re-purification via Ni-NTA to remove the glycosidase, and finally LCAT was polished via tandem Superdex 75 size exclusion chromatography (SEC) in reaction buffer (20 mM HEPES pH 7.5, 150 mM NaCl). The identity of the stable cell line expressing ΔNΔC-LCAT was initially verified by western blot with an anti-His antibody and abundant secretion into the conditioned media, followed by structural characterization of the correct protein.

Protein for biochemical analysis was made using pcDNA4 containing the codon-optimized human LCAT gene with a C-terminal 6x histidine-tag, which was transiently transfected in HEK293F cells as previously described (*Glukhova et al., 2015*). The cells were grown in suspension in FreeStyle medium supplemented with 100 U mL$^{-1}$ penicillin and 100 µg mL$^{-1}$ streptomycin, and conditioned media was harvested 5 d later. The secreted protein was purified via Ni-NTA and dialyzed against reaction buffer. The LCAT proteins used in pNPB, MUP, and DSF experiments were further polished via Superdex 75 s to remove any background contaminating reactivity.

## Crystallization and structure determination

ΔNΔC-LCAT was derivatized with isopropyl dodecyl fluorophosphonate (IDFP) to give ΔNΔC-IDFP in reaction buffer as previously described (*Manthei et al., 2017*). ΔNΔC-IDFP at 5 mg mL$^{-1}$ was incubated with 1 mM compound **1** for 30 min at room temperature in reaction buffer with 1% DMSO. Sparse matrix screens were set with a Crystal Gryphon (Art Robbins Instruments). Initial crystals of ΔNΔC-IDFP·**1** were obtained via sitting drop vapor diffusion from the Index HT screen. Crystals formed at 20 ˚C in a 1 µL drop with a protein to mother liquor ratio of 1:1. The crystals were optimized to a final condition of 0.25 M lithium sulfate, 0.1 M Tris pH 8.5, and 16% PEG 3350 via hanging drop vapor diffusion, and cryoprotected by moving the crystals to buffer with 0.2 M lithium sulfate, 0.1 M Tris pH 8.5, and 24% PEG 3350, and 20% glycerol. Crystals were frozen in nylon cryoloops (Hampton), and the data were collected at the Advanced Photon Source (APS) at Argonne National Laboratories on the LS-CAT 21-ID-G (λ = 0.97857) beam line. The data were processed and scaled with HKL-2000 (*Otwinowski and Minor, 1997*). The closed LCAT structure (PDB 5TXF) with the lid removed (residues 226–249) was used as a search model in molecular replacement with PHASER (*McCoy, 2007*) to generate initial phases. Non-crystallographic symmetry (NCS) restraints were applied to the two copies of LCAT per asymmetric unit during refinement in REFMAC5 (*Murshudov et al., 2011*) and Phenix (*Adams et al., 2010*) but removed during the final rounds of refinement. Reciprocal space refinement alternated with manual model building in Coot (*Emsley et al., 2010*). A Ni$^{2+}$ was observed coordinated by a portion of the exogenous His-tag beginning at residue 398 of chain A and aided in crystal packing. The final model was validated for stereochemical correctness with MolProbity (*Chen et al., 2010*).

## Soluble esterase assay

The esterase assay was performed as previously described (*Glukhova et al., 2015*) at least in triplicate. pNPB was diluted to 10 mM into reaction buffer containing 10% dimethylsulfoxide. The reaction was started by addition of 40 µL 1 µM LCAT containing either 3.2% DMSO or 11.1 µM compound **1** to 10 µL of pNPB. The increase in absorbance at 400 nm was monitored on a Spectramax plate reader for 15 min. Significance was determined using a one-way analysis of variance followed by Tukey's multiple comparisons post-test in GraphPad Prism.

## MUP hydrolysis assay

The lipase activity of LCAT was measured using MUP as a substrate. The assay was performed at room temperature in 0.1 M sodium phosphate buffer, pH 7.4 containing 0.01% Triton X-100. 4 µL of LCAT (6 nM final concentration) were dispensed into a 1536-well Greiner solid black plate. The same volume of assay buffer was dispensed into column 1 and 2 for a no-enzyme control. Then 23 nL DMSO or compounds titrated at 11-point 1:3 dilution series starting at 10 mM were transferred using a pintool. After 15 min incubation, 2 µL MUP (16 µM final concentration) was added to initiate the reaction. The hydrolysis of MUP was monitored using a ViewLux plate reader (excitation 380 nm/ emission 450 nm) for 20 min. The fluorescence signal was normalized against no-activator and no-enzyme control after subtraction of background signal (t = 0 min). To plot percent activation, in each assay 100% was set at the rate of LCAT or LCAT variant without compound. The resulting data were fitted to a sigmoidal dose response curve.

## Differential scanning fluorimetry

$T_m$ values were determined using an Applied Biosystems QuantStudio 7 Flex qPCR machine with two replicates performed at least in triplicate. LCAT at 0.05 mg mL$^{-1}$ was diluted into reaction buffer containing 5X Sypro Orange in a final volume of 10 µL in 384-well PCR plates. DMSO or

compound **1** was added so that all reactions contained 3% DMSO. The reactions were run from 25–95°C with a ramp rate of 0.03 °C s$^{-1}$. $T_m$ values were determined as the derivative using Protein Thermal Shift software. Significance was determined using a one-way analysis of variance followed by Tukey's multiple comparisons post-test in GraphPad Prism.

## MST binding assay

MST was used to determine the binding affinity of the compounds to LCAT. Recombinant proteins were labeled with a fluorophore using the Monolith His-tag labeling RED-Tris-NTA 2nd Generation kit following manufacturer's protocol. Compounds were titrated in a two-fold dilution series starting at 20 µM and incubated with the same volume of 100 nM labeled recombinant protein for 5 min at room temperature. Measurements were carried out in PBS containing 0.05% Tween-20 and standard capillaries using a Monolith NT.115 instrument (Nanotemper Technologies) with 50% LED excitation power, 60% MST power, MST on-time of 30 s and off-time of 5 s. $K_d$ values were calculated by fitting the thermophoresis signal at 20 s of the thermograph using MO.AffinityAnalysis software.

## Bio-Layer interferometry

A FortéBio Octet RED system was used to measure the binding of LCAT to ApoA-I HDLs. HDLs were prepared with 1,2-dipalmitoyl-sn-glycero-3-phosphocholine (DPPC), 1-palmitoyl-2-oleoyl-sn-glycero-3-phosphocholine (POPC), and 16:0 biotinyl Cap PE in a ratio of 49.5:49.5:1 (*Manthei et al., 2017*). HDLs were diluted 1/20 in assay buffer (1X PBS pH 7.4, 1 mM EDTA, 60 µM fatty acid free bovine serum albumin) and then immobilized on streptavidin tips for 600 s, followed by a wash in assay buffer for 600 s to remove unbound HDLs. The tips were then moved to buffer containing DMSO or compound **1** and allowed to equilibrate for 120 s before a baseline was established for 30 s. The tips were then moved into LCAT protein in assay buffer (containing DMSO or 10 µM **1**) or buffer alone (with DMSO or 10 µM **1**) as a control and allowed to associate for 200 s, and then dissociated in assay buffer for 480 s. All steps were performed at 25 °C with shaking at 1000 rpm. LCAT was titrated from 0.4 to 2.4 µM in triplicate. However, for some data sets (R244H, R244H + **1**, and Y51S/G71I), the 0.4 µM point was excluded due to low signal. The appropriate control of buffer containing DMSO or compound **1** was used to subtract the baseline and correct for drift using FortéBio's Data Analysis 7.0. The association and dissociation curves were fit using GraphPad Prism with a two-phase model. In order to determine $K_d$ values, the $k_{obs}$ (from association) were determined at each concentration for the fast phase and then plotted against LCAT concentration. The slope of the line was evaluated as $k_{on}$ using the equation $k_{obs} = k_{on}[LCAT]+k_{off}$ and the resultant $K_d = k_{off}/k_{on}$. For statistical analysis, the $k_{on}$, $k_{off}$, and $K_d$ for each replicate was determined individually and the results were compared to WT using a one-way analysis of variance followed by Tukey's multiple comparisons post-test in GraphPad Prism.

## DHE acyltransferase assay

Peptide-based HDLs were used in this assay as there is no difference between peptide HDLs and ApoA-I HDLs in both HDL binding and acyltransferase assays (*Manthei et al., 2017*). The peptide HDLs were made using the ESP24218 peptide with the sequence PVLDLFRELLNELLEALKQKLK (*Dassuex et al., 1999*; *Li et al., 2015*) with a DPPC:POPC:DHE ratio of 47:47:6 as previously described (*Manthei et al., 2017*). The assay was performed in 384-well low volume black microplates (Corning 4514) with a total assay volume of 16 µL. In each reaction, LCAT was diluted in assay buffer to 15 µg mL$^{-1}$ in the presence of either 1% DMSO or 10 µM **2** with 1% DMSO. Compound **2** was used in this assay because it has lower background fluorescence than **1**. The DHE HDLs were diluted in 1X PBS with 1 mM EDTA and 5 mM β-mercaptoethanol. 8 µL of the HDLs were added to the plate, and the reactions were initiated with 8 µL of LCAT, so that LCAT was assayed at 7.5 µg mL$^{-1}$ with and without 5 µM compound with a range of DHE concentrations from 0 to 50 µM. The reactions were stopped after 25 min at 37 °C with the addition of 4 µL of stop solution (1X PBS with 1 mM EDTA, 5 U mL$^{-1}$ cholesterol oxidase (COx), and 7% Triton X100). Following the addition of stop solution, the plates were incubated for another 60 min at 37 °C to allow for the COx to react. After the plates were re-equilibrated at room temperature, fluorescence was determined on a SpectraMax plate reader with excitation at 325 nm and emission at 425 nm, with a 420 nm cutoff. Reactions without LCAT were used for background subtraction, and reactions without LCAT and stop

solution lacking COx were used to generate a standard curve for DHE. Reactions were performed in triplicate with three independent experiments per LCAT variant. Data were processed via background subtraction to remove excess fluorescence that results from the higher concentrations of DHE. These values were divided by the slope of the line from the standard curve, which yields the amount of DHE-ester that resulted in each well, and then by time to determine the rate. Outliers were removed using automatic outlier elimination within Prism. For statistical analysis, the $V_{max}$ for each variant was compared to WT using a one-way analysis of variance followed by Tukey's multiple comparisons post-test in GraphPad Prism.

To determine $EC_{50}$ values, compound **2** was titrated from 0.004 to 10 µM, and the DHE concentration was set at 50 µM. LCAT was diluted in assay buffer and compound **2** dilutions were made with assay buffer containing 5.3% DMSO. 1.5 µL compound was added, then 6.5 µL LCAT, followed by 8 µL DHE. Dilutions were adjusted so that LCAT was assayed at 7.5 µg mL$^{-1}$, as above. All values were background subtracted to buffer with the same concentration of compound **2**. A standard curve was included in one experiment with DHE from 0 to 50 µM in order to adjust the final fluorescence values to a rate by dividing by the slope of the line and time (25 min), as above. Outliers were removed using automatic outlier elimination within Prism. For statistical analysis, the $EC_{50}$ for each variant was compared to WT using a one-way analysis of variance followed by Tukey's multiple comparisons post-test in GraphPad Prism.

### Statistical analysis

In most cases and as indicated in the methods and figure legends, statistical analysis was performed a one-way analysis of variance followed by Tukey's multiple comparisons post-test in GraphPad Prism. A paired t-test was used to compare the basal MUP hydrolysis levels. The statistical parameters, P value cutoffs, and number of replicates for each experiment are indicated in the table that corresponds to each experiment, the figure legends, and/or methods.

### Chemical synthesis

#### General methods for chemistry

All air or moisture sensitive reactions were performed under positive pressure of nitrogen with oven-dried glassware. Chemical reagents and anhydrous solvents were obtained from commercial sources and used as is. Preparative purification was performed on a Waters semi-preparative HPLC. The column used was a Phenomenex Luna C18 (5 micron, 30 × 75 mm) at a flow rate of 45 mL min$^{-1}$. The mobile phase consisted of acetonitrile and water (each containing 0.1% trifluoroacetic acid). A gradient of 10% to 50% acetonitrile over 8 min was used during the purification. Fraction collection was triggered by UV detection (220 nm). Analytical analysis for purity was determined by two different methods denoted as Final QC Methods 1 and 2. Method 1: analysis was performed on an Agilent 1290 Infinity Series HPLC with a 3 min gradient from 4% to 100% acetonitrile (containing 0.05% trifluoroacetic acid) followed by 1.5 min at 100% acetonitrile with a flow rate of 0.8 mL min$^{-1}$. A Phenomenex Luna C18 column (3 micron, 3 × 75 mm) was used at a temperature of 50°C. Method 2: analysis was performed on an Agilent 1260 with a 7 min gradient of 4% to 100% acetonitrile (containing 0.025% trifluoroacetic acid) in water (containing 0.05% trifluoroacetic acid) over 8 min run time at a flow rate of 1 mL min$^{-1}$. A Phenomenex Luna C18 column (3 micron, 3 × 75 mm) was used at a temperature of 50°C. Purity determination was performed using an Agilent Diode Array Detector for both Method 1 and Method 2. Mass determination was performed using an Agilent 6130 mass spectrometer with electrospray ionization in the positive mode. All of the analogs for assay have purity greater than 95% based on both analytical methods. $^{1}$H NMR spectra were recorded on Varian 400 MHz spectrometers.

The LCAT activators were synthesized as shown in the scheme in *Figure 7*.

#### Synthesis of 4-Hydroxy-3-(piperidin-4-yl)−4-(trifluoromethyl)−4,5-dihydro-1H-pyrazolo[3,4-*b*]pyridin-6(7H)-one, HCl (5)

Step 1: To a solution of tert-butyl 4-(5-amino-1H-pyrazol-3-yl)piperidine-1-carboxylate (**5a**, 799 mg, 3 mmol) in acetic acid (9 ml) was added ethyl 4,4,4-trifluoro-3-oxobutanoate (1657 mg, 9.0 mmol). The mixture was then heated at 60°C for 3 hr. After cooling to room temperature (RT), the mixture was diluted with EtOAc (20 mL) and was added saturated NaHCO$_{3(aq)}$ slowly until the pH of aqueous

**Figure 7.** Synthesis of piperidinylpyrazolopyridine and related compounds. Reagents and conditions: (**a**) ethyl 4,4,4-trifluoro-3-oxobutanoate, AcOH, 60°C, 3 hr, 57%. (**b**) HCl (4 M in 1,4-dioxane), 1,4-dioxane, 0°C to RT, 2 hr, **5** (96%), **6** (96%), **7** (83%). (**c**) 6-chloro-4-(trifluoromethyl)nicotinonitrile, Et₃N, EtOH, RT, 1 hr, 13%. (**d**) 2-chloro-5-(trifluoromethyl)pyrazine, (i-Pr)₂NEt, DMSO, RT, 3 hr, **2** (76%), **3** (76%), **8** (~7%). (**e**) tert-butyl 4-((methylsulfonyl)oxy) piperidine-1-carboxylate, K₂CO₃, DMF, 110°C, overnight, 38%. (**f**) H₂ balloon, cat. Pd/C, EtOH, RT, 2.5 hr; then ethyl 4,4,4-trifluoro-3-oxobutanoate, EtOH/AcOH (~1:2), 65–70°C, 2.5 hr, 86%. (**g**) hydroxylamine HCl salt, Et₃N, CH₂Cl₂, sealed, 55°C, overnight, 88%. (**h**) ethyl 4,4,4-trifluoro-3-oxobutanoate, EtOH/AcOH (~1:2), 70°C, 6 hr, 64%. (**i**) 2-chloro-5-(trifluoromethyl)pyrazine, Et₃N, DMF, RT, 3 hr, 47%.
DOI: https://doi.org/10.7554/eLife.41604.026

layer is ~7. The solution was extracted with EtOAc (50 mL x 3). The combined organic layer was dried (Na₂SO₄) and filtered. After removal of solvent, the product was purified by silica gel chromatography using 0–5% MeOH/EtOAc as the eluent to give tert-butyl 4-(4-hydroxy-6-oxo-4-(trifluoromethyl)−4,5,6,7-tetrahydro-1H-pyrazolo[3,4-b]pyridin-3-yl)piperidine-1-carboxylate (**5b**, 690 mg, 1.71 mmol, 56.9% yield).

Step 2: To a solution of tert-butyl 4-(4-hydroxy-6-oxo-4-(trifluoromethyl)−4,5,6,7-tetrahydro-1H-pyrazolo[3,4-b]pyridin-3-yl)piperidine-1-carboxylate (**5b**, 690 mg, 1.71 mmol) in 1,4-dioxane (4 ml) was added HCl (4M in dioxane, 2.6 mL, 10.2 mmol, 6 equivalents) at 0°C. The mixture was then stirred at RT for 2 hr. Then, hexane (15 mL) was added. The solid was filtered, washed with hexane (3 mL x 2), and then dried in vacuo to give 4-hydroxy-3-(piperidin-4-yl)−4-(trifluoromethyl)−4,5-dihydro-1H-pyrazolo[3,4-b]pyridin-6(7H)-one, HCl (**5**, 559 mg, 1.64 mmol, 96%). The material was used without further purification. LC-MS (Method 1): $t_R$ = 2.14 min, m/z (M + H)⁺ = 305.

### Synthesis of 7-Hydroxy-1-(piperidin-4-yl)−7-(trifluoromethyl)−6,7-dihydro-1H-imidazo[4,5-b]pyridin-5(4H)-one, HCl (6)

Step 1: To a mixture of 4-nitro-1H-imidazole (**6a**, 3.39 g, 30.0 mmol) and K₂CO₃ (4.2 g, 30.0 mmol) was added DMF (40 ml). The mixture was stirred at 110°C for 1 hr and tert-butyl 4-((methylsulfonyl)

oxy)piperidine-1-carboxylate (5.6 g, 20 mmol) was added and stirred at 110°C for overnight. The mixture was poured into EtOAc (200 mL)/H$_2$O (200 mL). The aqueous layer was extracted with EtOAc (150 mL x 2). The combined organic layer was concentrated to ~200 ml of solvent left. The organic solution was washed with H$_2$O (200 mL x 2), dried (Na$_2$SO$_4$) and filtered. After removal of solvent, some solid (nitroimidazole) from crude mixture can be filtered out by trituration with 50% EtOAc/hexane. The filtrate was concentrated and purified by silica gel chromatography using 30-70–100% EtOAc/hexane as the eluent to give tert-butyl 4-(4-nitro-1H-imidazol-1-yl)piperidine-1-carboxylate (**6b**, 2.25 g, 7.59 mmol, 38.0% yield).

Step 2: In a 2-neck flask was placed tert-butyl 4-(4-nitro-1H-imidazol-1-yl)piperidine-1-carboxylate (**6b**, 2.4 g, 8 mmol) and Pd-C (0.43 g, 0.40 mmol). Then, EtOH (50 ml) was added. The air was removed by house vacuum and refilled with N$_2$ for two times. Then, a H$_2$ balloon was attached. The N$_2$ air was removed by house vacuum and refilled with H$_2$ for three times. The mixture was stirred at RT for 2.5 hr. The H$_2$ balloon was removed and refilled with N$_2$. The mixture was filtered to remove most of Pd and the filtrate was then filtered again through a nylon 0.45 μM filter using EtOH as the eluent. The filtrate was concentrated to move most of EtOH until ~2–3 mL left. Then, to the crude product was added EtOH (6 mL), acetic acid (12 ml), and then ethyl 4,4,4-trifluoro-3-oxobutanoate (3.51 ml, 24.0 mmol). The mixture was then stirred at 65–70°C for 2.5 hr. After cooling to RT, the mixture was diluted with EtOAc (50 mL)/H$_2$O (30 mL) and was added saturated NaHCO$_{3(aq)}$ slowly until the pH of aqueous layer is ~7. The solution was extracted with EtOAc (70 mL x 3). The combined organic layer was dried (Na$_2$SO$_4$) and filtered. After removal of solvent, the product was purified by silica gel chromatography using 0-5–10% MeOH/EtOAc as the eluent to give tert-butyl 4-(7-hydroxy-5-oxo-7-(trifluoromethyl)−4,5,6,7-tetrahydro-1H-imidazo[4,5-b]pyridin-1-yl)piperidine-1-carboxylate (**6c**, 2.78 g, 6.87 mmol, 86% yield). LC-MS (Method 1): $t_R$ = 2.14 min, m/z (M + H)$^+$ = 405.

Step 3: To a solution of tert-butyl 4-(7-hydroxy-5-oxo-7-(trifluoromethyl)−4,5,6,7-tetrahydro-1H-imidazo[4,5-b]pyridin-1-yl)piperidine-1-carboxylate (**6c**, 222 mg, 0.549 mmol) in 1,4-dioxane (2 ml) was added HCl (4M in dioxane, 1.1 mL, 4.39 mmol, 8 equivalents) at 0°C. The mixture was then stirred at RT for 2 hr. Then, hexane (15 mL) was added, stirred, and then the hexane solvent was carefully removed (three times). The solid was then dried in vacuo to give 7-hydroxy-1-(piperidin-4-yl)−7-(trifluoromethyl)−6,7-dihydro-1H-imidazo[4,5-b]pyridin-5(4H)-one,  HCl (**6**, 180 mg, 0.528 mmol, 96% yield). The material was used without further purification. LC-MS (Method 1): $t_R$ = 2.07 min, m/z (M + H)$^+$ = 305.

## Synthesis of 4-Hydroxy-3-(piperidin-4-yl)−4-(trifluoromethyl)−4,5-dihydroisoxazolo[5,4-b]pyridin-6(7H)-one, HCl (7)

Step 1: To a mixture of tert-butyl 4-(2-cyanoacetyl)piperidine-1-carboxylate (**7a**, 2.02 g, 8 mmol) and hydroxylamine, HCl (0.70 g, 10.0 mmol) in CH$_2$Cl$_2$ (20 ml) was added Et$_3$N (2.23 ml, 16.0 mmol). The mixture was sealed and stirred at 55°C for overnight. After cooling to RT, the mixture was poured into CH$_2$Cl$_2$/H$_2$O (30 mL/30 mL). The aqueous layer was extracted with CH$_2$Cl$_2$ (30 mL). The combined organic layer was dried (Na$_2$SO$_4$) and filtered. After removal of solvent, the product was purified by silica gel chromatography using 60–100% EtOAc/hexane as the eluent to give tert-butyl 4-(5-aminoisoxazol-3-yl)piperidine-1-carboxylate (**7b**, 1.89 g, 7.08 mmol, 88% yield) $^1$H NMR (400 MHz, DMSO-$d_6$) δ 6.47 (s, 2H), 4.81 (s, 1H), 3.91 (d, J = 13.1 Hz, 2H), 2.79 br (s, 2H), 2.61 (tt, J = 11.5, 3.7 Hz, 1H), 1.80–1.69 (m, 2H), 1.43–1.33 (m, 11H); LC-MS (Method 1): $t_R$ = 3.05 min, m/z (M + Na)$^+$ = 290.

Step 2: To a solution of tert-butyl 4-(5-aminoisoxazol-3-yl)piperidine-1-carboxylate (**7b**, 535 mg, 2 mmol) in EtOH (2 ml) and AcOH (4 ml) was added ethyl 4,4,4-trifluoro-3-oxobutanoate (1105 mg, 6.0 mmol). The tube was sealed and heated at 70°C for 6 hr. The mixture was diluted with EtOAc/H$_2$O (10 mL/10 ml). Then, saturated NaHCO$_{3(aq)}$ was added dropwise to the stirring mixture until the pH of aqueous layer was ~7. The aqueous layer was extracted with EtOAc (30 mL x 2). The combined organic layer was dried (Na$_2$SO$_4$) and filtered. After removal of solvent, the product was purified by silica gel chromatography using 20–70% EtOAc/hexane as the eluent to give tert-butyl 4-(4-hydroxy-6-oxo-4-(trifluoromethyl)−4,5,6,7-tetrahydroisoxazolo[5,4-b]pyridin-3-yl)piperidine-1-carboxylate (**7c**, 520 mg, 1.28 mmol, 64.1% yield). $^1$H NMR (400 MHz, DMSO-$d_6$) δ 11.96 (s, 1H), 7.15 (s, 1H), 4.07–3.78 (m, 2H), 3.10 (d, J = 16.0 Hz, 1H), 2.97 (ddd, J = 11.5, 8.0, 3.6 Hz, 1H), 2.84 (d, J = 16.8 Hz,

1H), 2.78 (br s, 2H), 2.00 (d, $J$ = 12.8 Hz, 1H), 1.77 (ddd, $J$ = 13.4, 3.9, 1.9 Hz, 1H), 1.66–1.50 (m, 1H), 1.46–1.40 (m, 1H), 1.38 (s, 9H).

Step 3: To a solution of tert-butyl 4-(4-hydroxy-6-oxo-4-(trifluoromethyl)–4,5,6,7-tetrahydroisoxazolo[5,4-b]pyridin-3-yl)piperidine-1-carboxylate (7c, 520 mg, 1.28 mmol) in $CH_2Cl_2$ (5 ml) was added HCl (4M in dioxane, 10.3 mmol, 2.56 mL, ca. 8 equivalents). The mixture was stirred at RT for 2 hr. Then, hexane (15 mL) was added, stirred, and then the hexane solvent was carefully removed (three times). The solid was then dried in vacuo to give 4-hydroxy-3-(piperidin-4-yl)–4-(trifluoromethyl)–4,5-dihydroisoxazolo[5,4-b]pyridin-6(7H)-one, HCl (7, 366 mg, 1.07 mmol, 83% yield). The product was used without further purification. LC-MS (Method 1): $t_R$ = 2.28 min, m/z (M + H)$^+$ = 306.

## Synthesis of 6-(4-(4-Hydroxy-6-oxo-4-(trifluoromethyl)–4,5,6,7-tetrahydro-1H-pyrazolo[3,4-b]pyridin-3-yl)piperidin-1-yl)–4-(trifluoromethyl)nicotinonitrile, TFA (1)

To a solution of 4-hydroxy-3-(piperidin-4-yl)–4-(trifluoromethyl)–4,5-dihydro-1H-pyrazolo[3,4-b]pyridin-6(7H)-one, HCl (5, 34.1 mg, 0.1 mmol) in EtOH (2 mL) was added 6-chloro-4-(trifluoromethyl)nicotinonitrile (41.3 mg, 0.20 mmol) and Et₃N (0.042 mL, 0.30 mmol). The mixture was stirred at RT for 1 hr and then concentrated to remove most of EtOH. The mixture was dissolved in DMF, filtered through a filter and then submitted for purification by semi-preparative HPLC to give 6-(4-(4-hydroxy-6-oxo-4-(trifluoromethyl)–4,5,6,7-tetrahydro-1H-pyrazolo[3,4-b]pyridin-3-yl)piperidin-1-yl)–4-(trifluoromethyl)nicotinonitrile, TFA (1, 7.7 mg, 0.013 mmol, 13.1% yield). $^1$H NMR (400 MHz, DMSO-$d_6$) δ 12.14 (s, 1H), 10.46 (s, 1H), 8.69 (s, 1H), 7.29 (s, 1H), 6.72 (s, 1H), 4.68 (s, 2H), 3.40–3.29 (m, 1H), 3.05 (t, $J$ = 12.9 Hz, 2H), 2.87 (d, $J$ = 16.7 Hz, 1H), 2.70 (d, $J$ = 16.5 Hz, 1H), 1.91 (d, $J$ = 11.7 Hz, 1H), 1.73 (d, $J$ = 5.2 Hz, 2H), 1.64 (qd, $J$ = 12.5, 3.8 Hz, 1H); LC-MS (Method 2): $t_R$ = 4.70 min, m/z (M + H)$^+$ = 475.

## Synthesis of 4-Hydroxy-4-(trifluoromethyl)–3-(1-(5-(trifluoromethyl)pyrazin-2-yl)piperidin-4-yl)–4,5-dihydro-1H-pyrazolo[3,4-b]pyridin-6(7H)-one (2)

To a solution of 4-hydroxy-3-(piperidin-4-yl)–4-(trifluoromethyl)–4,5-dihydro-1H-pyrazolo[3,4-b]pyridin-6(7H)-one, HCl (5, 153 mg, 0.45 mmol) in DMSO (2 mL) was added 2-chloro-5-(trifluoromethyl)pyrazine (123 mg, 0.675 mmol) and then Hunig's base (0.16 mL, 0.90 mmol). The mixture was stirred at RT for 3 hr. The mixture was diluted with EtOAc (30 mL), washed with $H_2O$ (30 mL x 2), dried (Na₂SO₄) and filtered. After removal of solvent, the product was purified by silica gel chromatography using 45–85% EtOAc/hexane as the eluent to give 4-hydroxy-4-(trifluoromethyl)–3-(1-(5-(trifluoromethyl)pyrazin-2-yl)piperidin-4-yl)–4,5-dihydro-1H-pyrazolo[3,4-b]pyridin-6(7H)-one (2, 155 mg, 0.344 mmol, 76% yield) as a white solid. $^1$H NMR (400 MHz, DMSO-$d_6$) δ 12.15 (s, 1H), 10.46 (s, 1H), 8.48–8.46 (m, 2H), 6.72 (s, 1H), 4.62–4.58 (m, 2H), 3.38–3.31 (m, 1H), 3.08–2.96 (m, 2H), 2.87 (d, $J$ = 16.6 Hz, 1H), 2.70 (d, $J$ = 16.6 Hz, 1H), 1.96–1.84 (m, 1H), 1.76–1.62 (m, 3H); LC-MS (Method 2): $t_R$ = 4.70 min, m/z (M + H)$^+$ = 451.

## Synthesis of 7-Hydroxy-7-(trifluoromethyl)–1-(1-(5-(trifluoromethyl)pyrazin-2-yl)piperidin-4-yl)–6,7-dihydro-1H-imidazo[4,5-b]pyridin-5(4H)-one (3) and 7-(trifluoromethyl)–1-(1-(5-(trifluoromethyl)pyrazin-2-yl)piperidin-4-yl)–1H-imidazo[4,5-b]pyridin-5(4H)-one (8)

To a solution of 7-hydroxy-1-(piperidin-4-yl)–7-(trifluoromethyl)–6,7-dihydro-1H-imidazo[4,5-b]pyridin-5(4H)-one, HCl (6, 477 mg, 1.4 mmol) in DMSO (2 mL) was added 2-chloro-5-(trifluoromethyl)pyrazine (511 mg, 2.80 mmol) and then Hunig's base (0.489 mL, 2.80 mmol). The mixture was stirred at RT for 3 hr. The mixture was diluted with EtOAc (30 mL), washed with $H_2O$ (30 mL x 2), dried (Na₂SO₄) and filtered. After removal of solvent, to the crude product was added $CH_2Cl_2$ (10 mL). The product was filtered and washed with $CH_2Cl_2$ (2 mL x 3) and dried to give product (315 mg). The filtrate containing some desired product was concentrated and purified by silica gel chromatography using 5–10% MeOH/$CH_2Cl_2$ to give 163 mg of product. Total, 478 mg of product was obtained. 7-hydroxy-7-(trifluoromethyl)–1-(1-(5-(trifluoromethyl)pyrazin-2-yl)piperidin-4-yl)–6,7-dihydro-1H-imidazo[4,5-b]pyridin-5(4H)-one (3, 478 mg, 1.061 mmol, 76% yield) $^1$H NMR (400 MHz, DMSO-$d_6$) δ 10.27 (s, 1H), 8.48 (m, 2H), 7.75 (s, 1H), 7.22 (s, 1H), 4.75–4.54 (m, 3H), 3.13–2.96 (m,

3H), 2.76 (d, $J$ = 16 Hz, 1H), 2.15 (d, $J$ = 12.3 Hz, 1H), 2.06 (qd, $J$ = 12.4, 4.0 Hz, 1H), 1.93 (d, $J$ = 12.0 Hz, 1H), 1.77 (qd, $J$ = 12.3, 4.1 Hz, 1H); LC-MS (Method 2): $t_R$ = 4.71 min, m/z (M + H)$^+$ = 451. Some elimination side product was also collected and re-purified by silica gel chromatography using 0-5–10% MeOH/CH$_2$Cl$_2$ as the eluent to give 7-(trifluoromethyl)−1-(1-(5-(trifluoromethyl)pyrazin-2-yl)piperidin-4-yl)−1H-imidazo[4,5-b]pyridin-5(4H)-one (**8**, 40 mg, 0.093 mmol, 6.6%). $^1$H NMR (400 MHz, DMSO-$d_6$) δ 11.89 (br s, 1H), 8.57 (s, 1H), 8.54–8.48 (m, 2H), 6.75 (s, 1H), 4.71 (d, $J$ = 13.4 Hz, 2H), 4.46 (q, $J$ = 7.3, 6.9 Hz, 1H), 3.12 (dt, $J$ = 14.3, 8.4 Hz, 2H), 2.16–1.95 (m, 4H); LC-MS (Method 2): $t_R$ = 5.02 min, m/z (M + H)$^+$ = 433.

## Synthesis of 4-hydroxy-4-(trifluoromethyl)−3-(1-(5-(trifluoromethyl)pyrazin-2-yl)piperidin-4-yl)−4,5-dihydroisoxazolo[5,4-b]pyridin-6(7H)-one (9)

To a solution of 4-hydroxy-3-(piperidin-4-yl)−4-(trifluoromethyl)−4,5-dihydroisoxazolo[5,4-b]pyridin-6 (7H)-one, HCl (**7**, 68 mg, 0.2 mmol) in DMF (1 mL) was added 2-chloro-5-(trifluoromethyl)pyrazine (73.0 mg, 0.40 mmol) and Et$_3$N (0.084 mL, 0.60 mmol). The mixture was stirred at RT for 3 hr. The mixture was dropped into vigorously stirred H$_2$O (40 mL). The solid was filtered, washed with H$_2$O (2 × 3 mL) and then dried to give ~95 mg of desired product, which is ca. 90–95% purity. The product was dissolved in CH$_2$Cl$_2$ and purified by silica gel chromatography using 35–70% EtOAc/hexane as the eluent to give 4-hydroxy-4-(trifluoromethyl)−3-(1-(5-(trifluoromethyl)pyrazin-2-yl)piperidin-4-yl)−4,5-dihydroisoxazolo[5,4-b]pyridin-6(7H)-one (**9**, 42 mg, 0.093 mmol, 46.5% yield). $^1$H NMR (400 MHz, DMSO-$d_6$) δ 11.99 (s, 1H), 8.51–8.32 (m, 2H), 7.21 (s, 1H), 4.50 (dd, $J$ = 15.5, 12.0 Hz, 2H), 3.23–3.05 (m, 4H), 2.86 (d, $J$ = 16.8 Hz, 1H), 2.16 (d, $J$ = 12.7 Hz, 1H), 1.92 (d, $J$ = 12.3 Hz, 1H), 1.82–1.66 (m, 1H), 1.63–1.48 (m, 1H); LC-MS (Method 2): $t_R$ = 5.16 min, m/z (M + H)$^+$ = 452.

## Acknowledgements

We thank D J Leahy (Johns Hopkins University School of Medicine) for the expression vector pProEX HT-EndoH. We also thank the members of the Tesmer lab for their critique of the manuscript.

## Additional information

### Funding

| Funder | Grant reference number | Author |
| --- | --- | --- |
| National Heart, Lung, and Blood Institute | | Alan T Remaley |
| National Center for Advancing Translational Sciences | Division of Preclinical Innovation | Ajit Jadhav |
| American Heart Association | 15POST24870001 | Kelly A Manthei |
| National Institutes of Health | F32HL131288 | Kelly A Manthei |
| American Heart Association | 16POST27760002 | Wenmin Yuan |
| National Institutes of Health | HL071818 | John JG Tesmer |
| National Institutes of Health | HL122416 | John JG Tesmer |
| American Heart Association | 13SDG17230049 | Anna Schwendeman |

The funders had no role in study design, data collection and interpretation, or the decision to submit the work for publication.

### Author contributions

Kelly A Manthei, Conceptualization, Formal analysis, Supervision, Funding acquisition, Validation, Investigation, Visualization, Methodology, Writing—original draft, Writing—review and editing; Shyh-Ming Yang, Conceptualization, Formal analysis, Validation, Investigation, Visualization, Methodology, Writing—original draft, Writing—review and editing; Bolormaa Baljinnyam, Formal analysis, Investigation, Methodology, Writing—review and editing; Louise Chang, Alisa Glukhova, Investigation, Methodology, Writing—review and editing; Wenmin Yuan, Resources, Investigation,

Methodology, Writing—review and editing; Lita A Freeman, Methodology, Writing—review and editing; David J Maloney, Supervision, Writing—review and editing; Anna Schwendeman, Supervision, Funding acquisition, Writing—review and editing; Alan T Remaley, Conceptualization, Funding acquisition, Project administration, Writing—review and editing; Ajit Jadhav, Conceptualization, Supervision, Funding acquisition, Methodology, Project administration, Writing—review and editing; John JG Tesmer, Conceptualization, Supervision, Funding acquisition, Validation, Methodology, Project administration, Writing—review and editing

### Author ORCIDs
Kelly A Manthei http://orcid.org/0000-0003-3874-8228
Shyh-Ming Yang http://orcid.org/0000-0003-1928-136X
Ajit Jadhav http://orcid.org/0000-0001-7955-1451
John JG Tesmer http://orcid.org/0000-0003-1125-3727

### Decision letter and Author response
Decision letter https://doi.org/10.7554/eLife.41604.031
Author response https://doi.org/10.7554/eLife.41604.032

## Additional files

### Supplementary files
• Transparent reporting form
DOI: https://doi.org/10.7554/eLife.41604.027

### Data availability
The atomic coordinates and structure factors for crystals of the ΔNΔC-IDFP·1 complex have been deposited in the PDB with accession code 6MVD.

The following dataset was generated:

| Author(s) | Year | Dataset title | Dataset URL | Database and Identifier |
| --- | --- | --- | --- | --- |
| Manthei KA, Chang L, Tesmer JJG | 2018 | Crystal structure of Lecithin: cholesterol acyltransferase (LCAT) in complex with isopropyl dodec-11-enylfluorophosphonate (IDFP) and a small molecule activator | http://www.rcsb.org/structure/6MVD | RCSB Protein Data Bank, 6MVD |

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
