## [Decision Letter]

Thank you for submitting your article "Molecular basis for activation of lecithin:cholesterol acyltransferase by a compound that increases HDL cholesterol" for consideration by *eLife*. Your article has been reviewed by three peer reviewers, one of whom is a member of our Board of Reviewing Editors, and the evaluation has been overseen by Michael Marletta as the Senior Editor. The reviewers have opted to remain anonymous.

The reviewers have discussed the reviews with one another and the Reviewing Editor has drafted this decision to help you prepare a revised submission.

Summary:

This comprehensive report uses structural and biochemical analysis to describe a compound that increases the activity of LCAT by ~4-fold by an interesting, indirect mechanism. LCAT is the key enzyme in the conversion of discoidal pre-β HDL to spherical α-HDL, and activators of LCAT hold promise for disease settings where LCAT is deficient (fish eye disease and familial LCAT deficiency) and for modulating HDL levels. The reviewers all agreed that the data were of high quality, the findings were significant, and recommended acceptance pending the addressing of a few concerns.

Essential revisions:

1) Please provide all materials used for structural determination: the electron density map, including the major structural elements and the loop regions (particularly, the lid) that are discussed in the manuscript should be presented in the supplementary materials. Also, the authors should provide the image to show the crystal packing in the supplementary materials.

2) The Ramachandran outliers are very high (over 1%). Although the reviewers understand that the protein contains many flexible regions that may be difficult to refine, the authors should refine the structure; also the authors should provide the molprobity score and clash score in Table 4.

3) Please provide a model cartoon figure of LCAT outlining the various domains of the enzyme, where the activator and IDFP bind, and their speculative mode of action of the activator? While all of these are provided in various figures, a single summary model figure would be very helpful for readers.

---

## [Author Response]

Essential revisions:1) Please provide all materials used for structural determination: the electron density map, including the major structural elements and the loop regions (particularly, the lid) that are discussed in the manuscript should be presented in the supplementary materials. Also, the authors should provide the image to show the crystal packing in the supplementary materials.

We have added a supplementary figure (new Figure 2—figure supplement 1) containing omit maps for the lid and the αA-αA′ loop for chain A. The crystal packing is also shown in this figure. Furthermore, the structure factors have already been deposited and released at the PDB (code 6DTJ).

2) The Ramachandran outliers are very high (over 1%). Although the reviewers understand that the protein contains many flexible regions that may be difficult to refine, the authors should refine the structure; also the authors should provide the molprobity score and clash score in Table 4.

We apologize for the high Ramachandran outliers, and we have refined the structure with these outliers in mind and the outliers are now 0.5%. We updated the table with these values and added the MolProbity score and clashscore. We also updated the PDB, the new PDB code will be 6MVD, which will replace 6DTJ and be released upon publication.

3) Please provide a model cartoon figure of LCAT outlining the various domains of the enzyme, where the activator and IDFP bind, and their speculative mode of action of the activator? While all of these are provided in various figures, a single summary model figure would be very helpful for readers.

We agree and added a model, see new Figure 6.